# Quantification of *Salmonella enterica* serovar Typhimurium population dynamics in murine infection using a highly diverse barcoded library

Julia A Hotinger[1,2†], Ian W Campbell[1,2†], Karthik Hullahalli[1,2], Akina Osaki[1,2], Matthew K Waldor[1,2,3]*

[1]Division of Infectious Diseases, Brigham & Women's Hospital, Boston, United States; [2]Department of Microbiology, Harvard Medical School, Boston, United States; [3]Howard Hughes Medical Institute, Boston, United States

*For correspondence:
mwaldor@research.bwh.harvard.edu

†These authors contributed equally to this work

Competing interest: The authors declare that no competing interests exist.

## eLife Assessment

This **important** study reports a detailed quantification of the population dynamics of *Salmonella enterica* serovar Typhimurium in mice. Bacterial burden and founding population sizes across various organs were quantified, revealing pathways of dissemination and reseeding of the gastrointestinal tract from systemic organs. Using various techniques, including genetic distance measurements, the authors present **compelling** evidence to support their conclusions, thus presenting new knowledge that will be of broad interest to scientists focusing on infectious diseases.

**Abstract** Murine models are often used to study the pathogenicity and dissemination of the enteric pathogen *Salmonella enterica* serovar Typhimurium. Here, we quantified *S.* Typhimurium population dynamics in mice using the STAMPR analytic pipeline and a highly diverse *S.* Typhimurium barcoded library containing ~55,000 unique strains distinguishable by genomic barcodes by enumerating *S.* Typhimurium founding populations and deciphering routes of spread in mice. We found that a severe bottleneck allowed only one in a million cells from an oral inoculum to establish a niche in the intestine. Furthermore, we observed compartmentalization of pathogen populations throughout the intestine, with few barcodes shared between intestinal segments and feces. This severe bottleneck widened and compartmentalization was reduced after streptomycin treatment, suggesting the microbiota plays a key role in restricting the pathogen's colonization and movement within the intestine. Additionally, there was minimal sharing between the intestine and extraintestinal organ populations, indicating dissemination to extraintestinal sites occurs rapidly, before substantial pathogen expansion in the intestine. Bypassing the intestinal bottleneck by inoculating mice via intravenous or intraperitoneal injection revealed that *Salmonella* re-enters the intestine after establishing niches in extraintestinal sites by at least two distinct pathways. One pathway results in a diverse intestinal population. The other re-seeding pathway is through the bile, where the pathogen is often clonal, leading to clonal intestinal populations and correlates with gallbladder pathology. Together, these findings deepen our understanding of *Salmonella* population dynamics.

## Introduction

*Salmonella enterica* is a foodborne pathogen that causes hundreds of millions of infections and over 150,000 deaths worldwide each year (*Lamichhane et al., 2024*). Its over 2000 serovars are divided

into human-restricted typhoidal serovars (i.e. Typhi) that cause typhoid fever, a disseminated bacterial infection, and non-typhoidal serovars (e.g. Typhimurium) that are typically limited to intestinal disease in humans. There is, however, a growing number of extraintestinal infections caused by non-typhoidal serovars (*Worley, 2023*).

*Salmonella* is an extremely versatile pathogen capable of replicating both outside of and within host cells in a variety of animal hosts (*Ruby et al., 2012*; *Coombes et al., 2005*; *Higginson et al., 2016*). *Salmonella* has two primary pathogenicity islands (SPI-1 and SPI-2), which facilitate infection in humans and are required for colonization and disease in mice (*Fierer et al., 2012*; *Li, 2022*; *Zhang et al., 2018*), one of the most commonly used model hosts for studying *Salmonella* pathogenesis (*Xu and Hsu, 1992*). In C57BL/6 J mice, *Salmonella enterica* serovar Typhimurium (*S.* Typhimurium) causes a disseminated infection. This murine model has helped to reveal that the pathogen employs numerous virulence factors to drive multiple interconnected mechanisms for escaping from the intestine to reach systemic organs.

Identifying changes in the frequency of genomic barcodes in otherwise identical bacteria throughout infection is a powerful tool for monitoring pathogen population dynamics in experimental models of infection. However, previous studies using barcoded *Salmonella* to study population dynamics during dissemination were limited in their resolution because they used a small number of barcodes (<30 unique barcodes; *Kaiser et al., 2013*; *Kaiser et al., 2014*; *Dybowski et al., 2015*; *Lam and Monack, 2014*; *Grant et al., 2008*; *Dybowski et al., 2017*), hampering their capacity to measure bottlenecks, the host barriers to infection. These advances notwithstanding, we lack a granular understanding of *Salmonella* population dynamics in murine hosts that can be provided by evaluating the quantity, diversity, and similarity of barcoded *Salmonella* populations across many organs.

Inspired by previous work, we created an *S.* Typhimurium library containing over 55,000 unique barcodes and used the STAMPR (Sequence Tag-based Analysis of Microbial Populations in R) analytical framework (*Hullahalli et al., 2021*; *Abel et al., 2015*) to quantify the host bottlenecks restricting *Salmonella* colonization and dissemination. STAMPR enables the calculation of the size of the founding population, which represents the number of bacterial cells from the inoculum that survive the bottleneck and give rise to the observed population. In addition, the STAMPR analysis pipeline can determine the relative similarity of bacterial populations at different sites within the same animal by comparing the frequency and identity of barcodes. The quantification of both founding populations and similarity between populations can reveal new insights into host bottlenecks to infection and unexpected patterns of bacterial spread within the host (*Hullahalli et al., 2021*; *Abel et al., 2015*; *Holmes et al., 2025*; *Zhang et al., 2017*; *Chevée et al., 2024*; *Bachta et al., 2020*; *Campbell et al., 2023*).

Here, oral administration of the *S.* Typhimurium barcoded library revealed that a tight bottleneck restricts *S.* Typhimurium intestinal colonization. Furthermore, we observed compartmentalized subpopulations within the intestine that did not intermix or contribute to the pathogen population shed in the feces. Disrupting the microbiota prior to inoculation by pretreating mice with streptomycin significantly relaxed the severe intestinal colonization bottleneck and increased sharing between intestinal and fecal bacterial populations, demonstrating the importance of the microbiome in protecting against infection and indicating the microbiome has a role in restricting the movement of *S.* Typhimurium within the intestine. Comparing intestinal *S.* Typhimurium populations to disseminated populations in other tissues, we discovered that *S.* Typhimurium dissemination occurs without the need to establish a replicative niche in the intestine, consistent with a previously presented hypothesis (*Watson and Holden, 2010*). Bypassing the intestine by administering *S.* Typhimurium by intravenous (IV) or intraperitoneal (IP) injection nearly eliminated the bottleneck to colonizing extraintestinal sites, suggesting the primary bottleneck to colonizing these sites occurs within or while exiting the intestine. Furthermore, comparisons of the *S.* Typhimurium populations in the bile and intestine following IV and IP inoculation revealed that bile from the gallbladder is one source of bacteria for intestinal re-seeding, a phenomenon observed in humans with recurrent infections (*Crawford et al., 2010*; *Shrout, 2012*). These observations provide a strong foundation for further work defining the mechanisms and dynamics of *Salmonella* spread during infection.

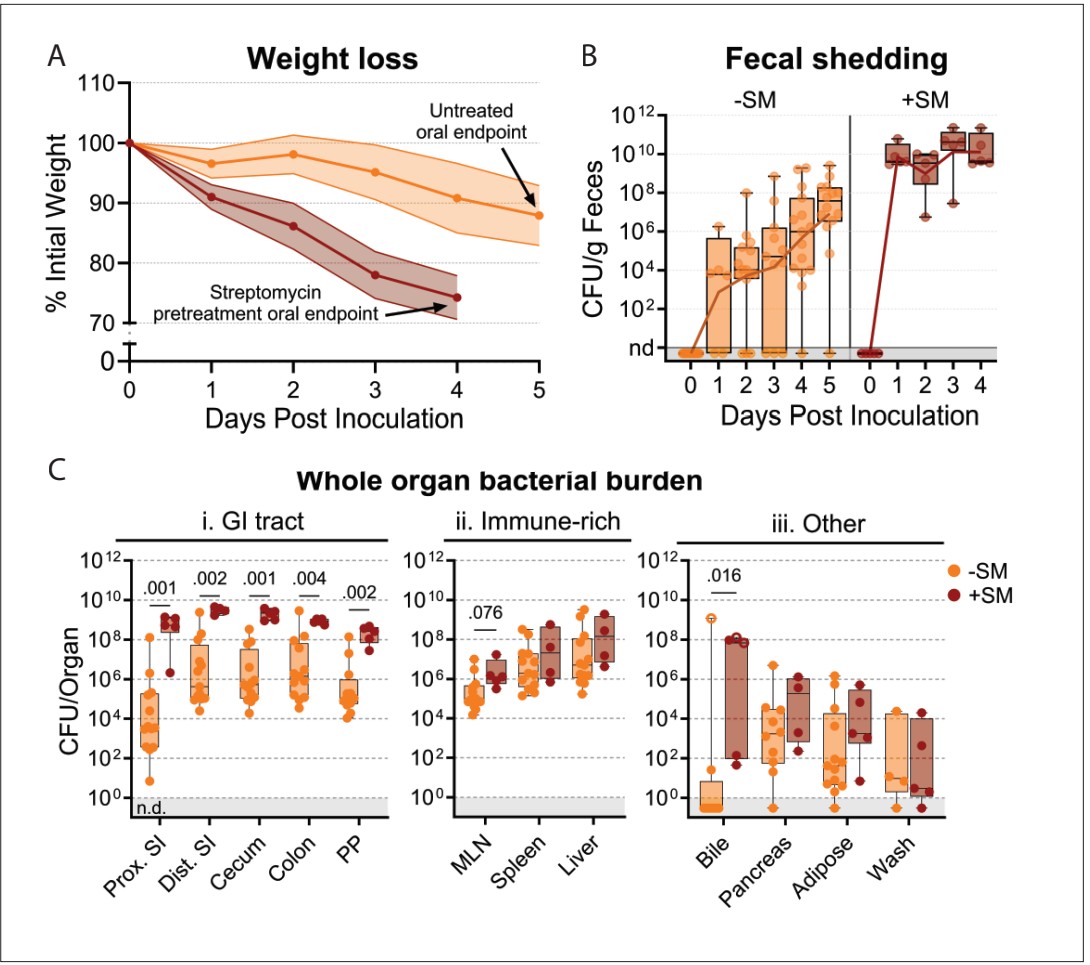

**Figure 1.** *S.* Typhimurium burden following orogastric inoculation of untreated or streptomycin-treated mice. (**A**) Percentage of initial weight over time; means and standard deviations are shown. (**B**) Bacterial fecal burden. Box and whisker plots represent max-to-min and interquartile ranges. (**C**) Bacterial burden in organs and fluids. Open circles indicate when the whole gallbladder was used instead of bile. Box and whisker plots represent max-to-min and interquartile ranges. Untreated n=16 (8 males and 8 females), streptomycin (SM) pretreated n=5 (female) unless otherwise noted. Sex-disaggregated data in *Figure 3—figure supplement 2*. Mann-Whitney tests were used for statistical analyses. Values with p<0.1 shown. Abbreviations: Adipose, left perigonadal adipose tissue; MLN, mesenteric lymph node; PP, Peyer's patches; SI, small intestine; wash, peritoneal wash.

The online version of this article includes the following figure supplement(s) for figure 1:

**Figure supplement 1.** Weight of cecum (**A**) and colon (**B**) after different infection schemes.

## Results

### Streptomycin treatment widens the bottleneck impeding *Salmonella* intestinal colonization

Streptomycin pretreatment is often used to sensitize mice to *S.* Typhimurium orogastric infection. This antibiotic reduces the microbiota and is thought to heighten the pathogen burden and consistency of intestinal colonization (*Bohnhoff et al., 1954*). We orogastrically administered $10^8$ colony-forming units (CFU) of a barcoded *S.* Typhimurium library into 8-week-old untreated and streptomycin-treated C57BL/6 J mice to investigate how streptomycin pretreatment modifies *S.* Typhimurium population dynamics. We monitored common markers of disease and colitis during infection (*Figure 1—figure supplement 1*). Consistent with previous reports (*Bohnhoff et al., 1954*; *Stecher et al., 2006*; *Barthel et al., 2003*), the streptomycin-treated animals lost weight more rapidly than animals that were not streptomycin-treated and were sacrificed on day 4 post-inoculation (dpi), after they had lost >20% of their body weight (*Figure 1A*). Fecal shedding of *S.* Typhimurium gradually increased in untreated

mice. In contrast, animals pretreated with streptomycin rapidly and uniformly began shedding *S.* Typhimurium, with fecal burdens higher than the inoculum by 24 hr post-inoculation and sustained throughout the experiment (*Figure 1B*).

We measured *S.* Typhimurium burden in 12 organs/fluids at 4 (+SM) or 5 (untreated/-SM) days post-inoculation, while they were at the peak of disease. Organs/tissues were classified into three groups based on the pathogen burden and differences between the streptomycin-treated and untreated animals (*Figure 1Ci–iii*). The gastrointestinal (GI) tract organs, including the proximal and distal small intestine (SI), cecum, colon, and Peyer's patches (PP), are presumed to be the initial site of infection after orogastric inoculation and generally had 100–10,000-fold higher burden in streptomycin-treated animals (*Figure 1Ci*); also, there was much more variation in pathogen burdens within the GI tracts of untreated animals. In contrast, in the extraintestinal immune-rich organs, such as the mesenteric lymph node (MLN), spleen, and liver, the *S.* Typhimurium burdens and variation in burden were similar in the two groups (*Figure 1Cii*). In the remaining organs and fluids, including the bile, pancreas, perigonadal adipose tissue ('adipose'), and peritoneal wash ('wash'), bacterial burdens were generally lower than in other tissues, except for bile, and the burdens in treated and untreated animals were similar. Together, these observations reveal that although streptomycin treatment markedly elevates the *S.* Typhimurium burden in the GI tract, it does not appear to significantly alter the pathogen burden at sites beyond the intestine.

Counting the number of *S.* Typhimurium cells within an organ does not directly measure the bottleneck, the bacterial population encountered in establishing its niche because bacterial replication obscures the effects of bottlenecks (i.e. the observed population is the net outcome of a bottleneck followed by replication and migration from other sites). To directly measure the bottleneck impeding

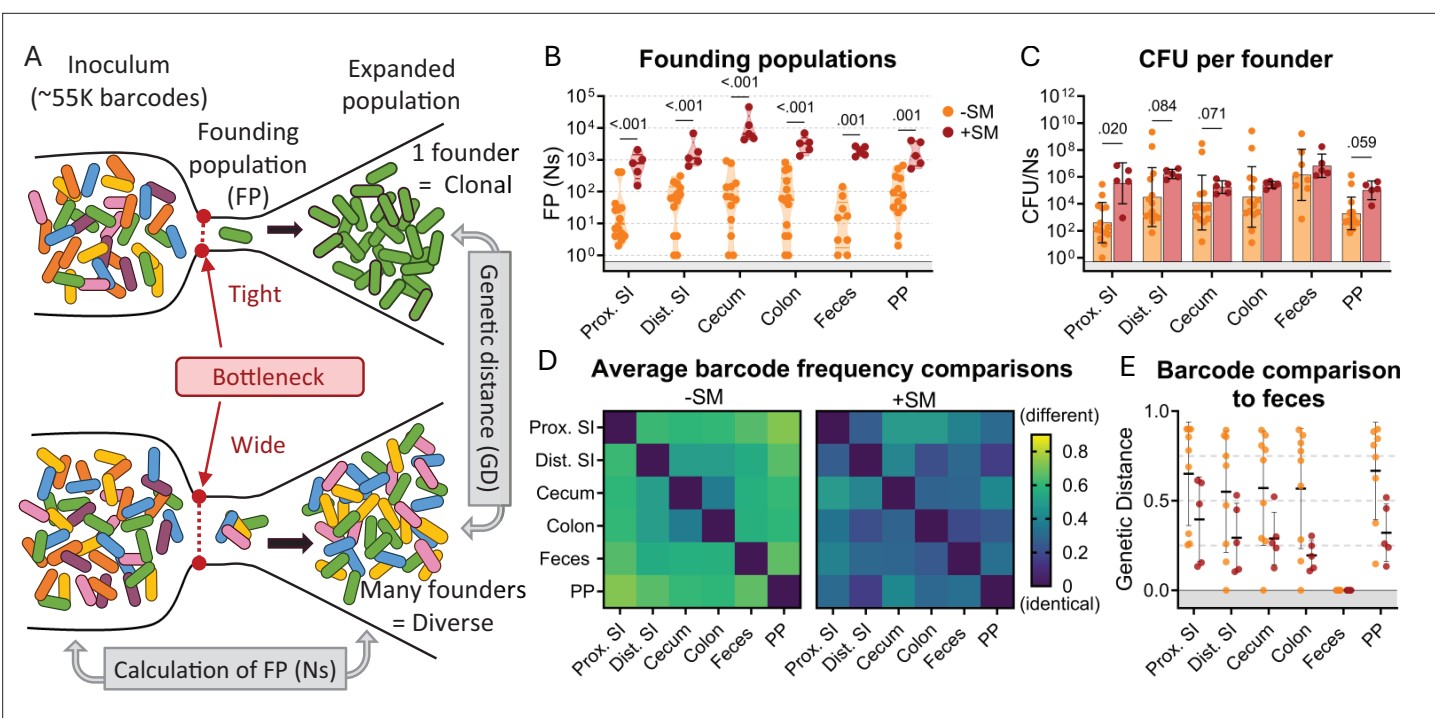

**Figure 2.** The bottleneck to *S.* Typhimurium intestinal colonization in orogastrically inoculated mice is widened after streptomycin treatment. (**A**) Schematic depicting the effect of a tight versus wide bottleneck on a diverse inoculum and the STAMPR analytical methods used to calculate the founding population (FP) and compare populations at separate sites of infection using genetic distance (GD) analysis. (**B**) Founding populations (Ns) of intestinal samples. Truncated violin plot with all points shown. (**C**) CFU per founder (CFU/Ns) in intestinal tissues. Bars are geometric means and geometric standard deviations. (**D**) Heatmaps of average genetic distance comparisons throughout the GI tract in untreated (left) and streptomycin-treated (right) mice. (**E**) Comparison of genetic distance of GI tract to fecal samples. Mann-Whitney tests were used for statistical analyses. Values with p<0.1 shown. Abbreviations: PP, Peyer's patches; SI, small intestine; SM, streptomycin.

The online version of this article includes the following figure supplement(s) for figure 2:

**Figure supplement 1.** *S.* Typhimurium barcoded library diversity and standard curve.

**Figure supplement 2.** Barcodes from different regions of the intestine and Peyer's patches from a single animal are largely distinct.

intestinal colonization, the experiments described above were carried out with genomically barcoded *S.* Typhimurium. We introduced neutral tags into our *S.* Typhimurium population by integrating short ~25 base pair sequences (barcodes) into the genome on a Tn7 transposon, resulting in a library of ~55,000 strains that are isogenic except for their barcodes (*Hullahalli et al., 2021*; *Abel et al., 2015*). These barcodes can be detected by amplicon sequencing and changes in the abundance and frequencies of barcodes in the library allow us to determine the number of unique cells from the inoculum that gave rise to the observed population, referred to as the 'founding population' or 'founders'. The number of founders is estimated using the STAMPR analytical pipeline by a metric called Ns, which employs a multinomial resampling strategy to determine the sampling depth required to observe a specific number of barcodes given the distribution of barcodes in the inoculum (*Hullahalli et al., 2021*). The size of the founding population in an organ and its barcode diversity reflects the bottleneck experienced by the inoculum, with tighter bottlenecks resulting in fewer unique barcodes than wider bottlenecks (*Figure 2A*). Before performing animal experiments, we determined the resolution of STAMPR analysis with our library by creating experimental bottlenecks in a culture of the *S.* Typhimurium library by plating serial dilutions and found that our calculations accurately predicted the number of colony-forming units (which can be considered the true size of the founding population) up to ~700,000 CFU (*Figure 2—figure supplement 1*).

There was considerable variation in *S.* Typhimurium founding population (Ns) sizes in untreated animals (*Figure 2B*), reflecting inter-animal variation in the bottleneck to intestinal colonization, which likely accounts for the variable timing and quantity of fecal shedding observed in this model (*Figure 1B*). In most parts of the intestine of untreated mice, there were only ~$10^2$ unique founders (*Figure 2B*), suggesting that at this dose ($10^8$ CFU) there is a severe intestinal colonization bottleneck that reduces the inoculum population by about ~$10^6$-fold. Streptomycin treatment widened this bottleneck considerably, increasing the founding population by 10–100-fold in all regions of the intestine (*Figure 2B*). This increased founding population following streptomycin treatment demonstrates that streptomycin clearance of the microbiota and its downstream consequences, such as inflammation (*Bohnhoff et al., 1954*), removes some of the barriers to *S.* Typhimurium establishing a replicative niche in the intestine.

Streptomycin treatment increased both the total *S.* Typhimurium burden (*Figure 1Ci*) and the number of founders (*Figure 2B*) in the intestine. Replication of founders generally accounts for the majority of the burden in most tissues. The average net expansion of the population can be measured by dividing bacterial burden by the founding population (CFU/Ns). This calculation revealed that there is generally more net expansion in streptomycin-treated versus untreated animals (*Figure 2C*); that is, the increase in CFU following streptomycin treatment is caused by both increased number of founders and increased net replication. These observations suggest that streptomycin treatment not only widens the bottleneck impeding *S.* Typhimurium colonization (increased founders) but also creates a more permissive niche for pathogen population expansion.

## Streptomycin treatment decreases compartmentalization of *S.* Typhimurium populations within the intestine

The STAMPR analytical pipeline enables comparison of barcoded bacterial populations sampled from different sites within the same animal. The similarity in the frequency of barcodes between organs is quantified with a metric of genetic distance (*Figure 2A*; *Abel et al., 2015*). Organs containing barcodes with indistinguishable proportions have a genetic distance (GD) of zero, while those that do not share barcodes have a genetic distance of nearly one (see Materials and methods). We compared the similarity of barcode frequencies in samples from different intestinal regions. Except for the cecum and colon, in untreated animals the *S.* Typhimurium populations in different regions of the intestine were dissimilar (Avg. GD ranged from 0.369 to 0.729, 2D left); that is, there is little sharing between populations in the intestine. These data suggest that there are separate bottlenecks in different regions of the intestine that cause stochastic differences in the identity of the founders. Interestingly, when these founders replicate, they do not mix, remaining compartmentalized with little sharing between populations throughout the intestinal tract (i.e. barcodes found in one region are not in other regions; *Figure 2—figure supplement 2*). This was surprising as the luminal contents, an environment presumably conducive to bacterial movement, were not removed from these samples. Further supporting this compartmentalized population hypothesis is the lack of barcode overlap between individual Peyer's

patches taken from the proximal, medial, and distal small intestine of an individual mouse (*Figure 2— figure supplement 2*). Streptomycin treatment increased the similarity in *S.* Typhimurium populations derived from different parts of the intestine (Avg. GD ranged from 0.157 to 0.407, *Figure 2D* right), suggesting that reduction of the microbiota through streptomycin pretreatment increases the mixing of the populations between the intestinal compartments.

S. Typhimurium is primarily transmitted through the feces. We attempted to identify the origin from within the intestinal tract of fecal *S.* Typhimurium through genetic distance comparisons of fecal samples to different regions of the intestine. Unexpectedly, in approximately half of the untreated animals, the fecal barcodes were highly dissimilar from the barcodes present in all regions of the intestine (GD >0.75; *Figure 2E*), indicating that the majority of the *S.* Typhimurium population at intestinal sites are not being shed into the feces. In contrast, mice treated with streptomycin had similar populations in intestinal and fecal samples (Avg. GD = 0.299 ± 0.06; *Figure 2D*), reflecting an increase in sharing between these populations. Thus, STAMPR-based analysis of the high-density barcoded *S.* Typhimurium library enabled quantitative assessments of intestinal bottlenecks and revealed that streptomycin treatment markedly alters the compartmentalization of *S.* Typhimurium replication in the intestine and the source of the pathogen shed in the feces.

## S. Typhimurium disseminates out of the intestine before establishing an intestinal replicative niche

In mice, *S.* Typhimurium routinely spreads out of the intestine to colonize extraintestinal organs. In the intestine, streptomycin treatment increased *S.* Typhimurium burdens by 100–10,000-fold relative to untreated animals. In contrast, streptomycin treatment did not markedly increase the burden of *S.* Typhimurium in systemic organs (*Figure 1Cii–iii*). Nevertheless, barcode analysis revealed that streptomycin treatment increased the size of the founding population in extraintestinal organs by ~10-fold (*Figure 3A*), suggesting that the net bottleneck to dissemination is relaxed by streptomycin treatment, likely contributing to the more rapid disease progression observed in streptomycin-treated animals (*Figure 1A*). However, since the founding population is calculated based on the abundance of barcodes in an organ sample at sacrifice, it is difficult to discern if streptomycin treatment modifies the net bottleneck to dissemination by relaxing the individual bottlenecks to reaching the intestine, colonizing the intestine, disseminating from the intestine, and/or establishing a niche in extraintestinal organs.

To gain further insight into when *S.* Typhimurium disseminates to extraintestinal sites, we compared the similarity of barcodes found in *S.* Typhimurium isolated from the liver to those in the intestine. The similarity between the initial site of infection (in this case the intestine) and secondary sites differentiates whether the secondary population arose soon after inoculation, before the population experienced the profound intestinal bottleneck ('early' in *Figure 3B*), or at a later point, after the initial bottlenecks and subsequent replication ('late' in *Figure 3B*; *Holmes et al., 2025*). We expect that early spread would yield dissimilar barcode frequencies between intestinal and extraintestinal sites, whereas, with late spread, there would be greater barcode similarity in the two populations. Genetic distance comparison of liver samples to other sites revealed that, regardless of streptomycin treatment, there was very little sharing of barcodes between the intestine and extraintestinal sites (Avg. GD >0.75, *Figure 3C*). Furthermore, the MLN and spleen populations also lacked similarity with the intestine (*Figure 3—figure supplement 1*). These analyses strongly support the idea that *S.* Typhimurium disseminates to extraintestinal organs relatively early following inoculation, before it establishes a replicative niche in the intestine.

It is possible that orogastric gavage can cause trauma to the esophagus that may result in the introduction of bacteria to the bloodstream during inoculation (*Nilsson et al., 2019*), potentially giving rise to the independent populations observed in the intestine and extraintestinal sites ('direct' in *Figure 3B*). To exclude the possibility that we unintentionally introduced the pathogen into circulation through orogastric gavage, streptomycin-treated mice were administered water containing $10^8$ CFU of the *S.* Typhimurium barcoded library in their oral cavity and allowed to drink (+SM, drinking). These animals showed highly similar weight loss and fecal shedding patterns as observed in streptomycin-treated mice orogastrically gavaged (+SM, Gavage) with the same dose of *S.* Typhimurium (*Figure 3— figure supplement 3*). Furthermore, the bacterial burdens in the intestine and extraintestinal organs were indistinguishable regardless of the method of oral inoculation (*Figure 3D*).

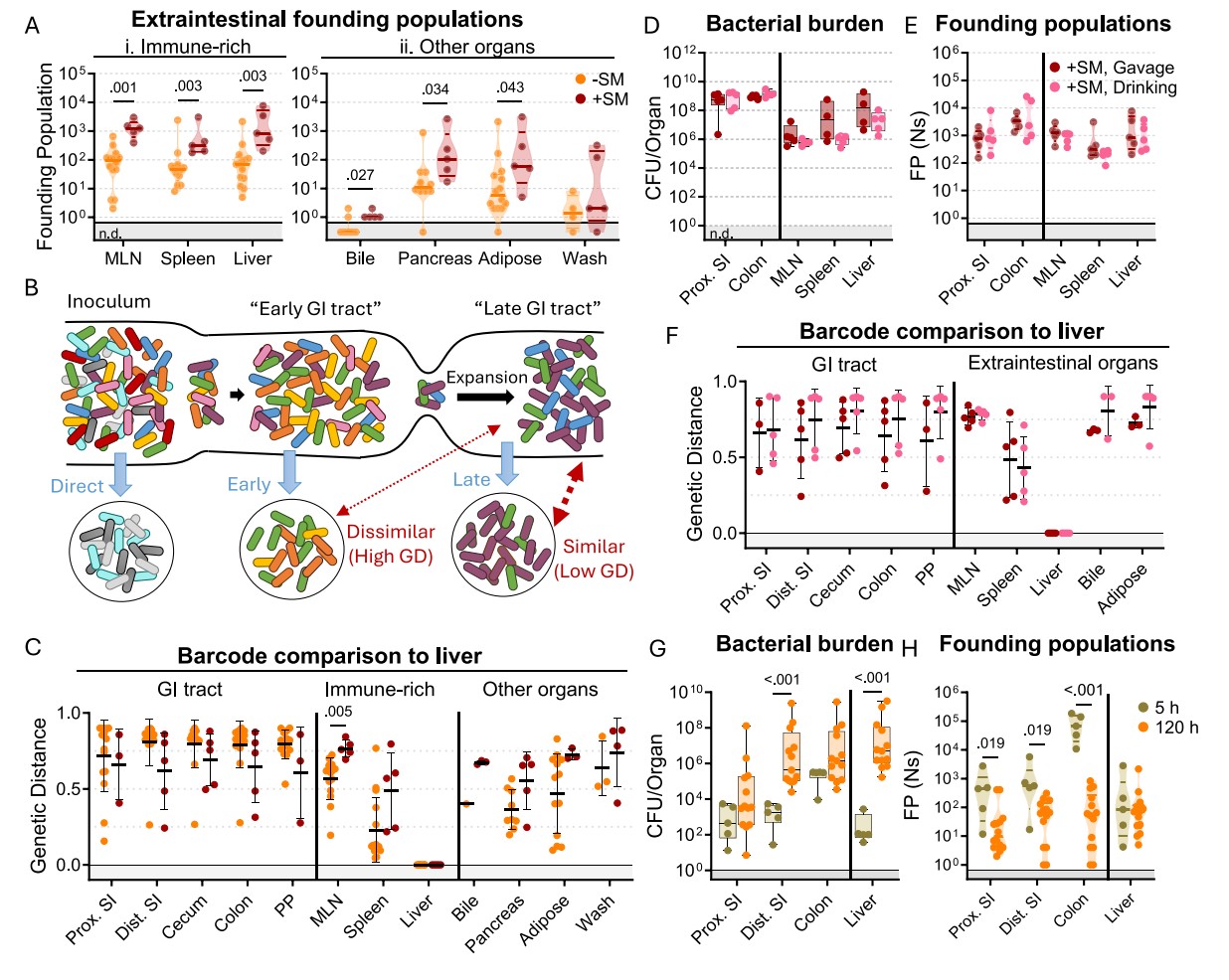

**Figure 3.** *S.* Typhimurium disseminates from the intestine before substantial replication. (**A**) Founding populations and (**C**) genetic distance from the liver after orogastric inoculation (Gavage) of *S.* Typhimurium with (+SM) or without (-SM) streptomycin pretreatment. (**B**) Scheme depicting proposed dissemination patterns during infection. (**D–F**) Organ CFUs (**D**), founding populations (**E**), and genetic distance from the liver (**F**) after inoculation via drinking (+SM, Drinking) are not statistically different from orogastric gavage (+SM, Gavage). Data for +SM, gavage mice repeated from *Figures 1–3*. (**G–H**) Organ CFUs (**G**) and founding populations (**H**) in untreated mice at 5 or 120 hr after inoculation. 120 hr data repeated from *Figures 1–3*. +SM, Drinking n=5, untreated 5 hr n=5. Mann-Whitney tests were used for statistical analyses. Values with p<0.1 shown. Abbreviations: GD, genetic distance; GI, gastrointestinal; MLN, mesenteric lymph node; PP, Peyer's patches; SI, small intestine.

The online version of this article includes the following figure supplement(s) for figure 3:

**Figure supplement 1.** *S.* Typhimurium disseminates to the MLN and spleen prior to substantial replication in the intestine.

**Figure supplement 2.** Sex-disaggregated data in mice inoculated through orogastric gavage.

**Figure supplement 3.** Extended data for streptomycin pretreated mice inoculated through orogastric gavage and drinking.

Drinking and orogastric gavage also yielded very similar founding population sizes in all organs (*Figure 3E*). The similarity in number of founders suggests that the two inoculums experienced indistinguishable bottlenecks, an unlikely event if orogastric gavage led to routine bloodstream inoculation. Moreover, changing oral administration from orogastric gavage to drinking did not impact the lack of similarity between *S.* Typhimurium barcodes present in liver samples and other organs (*Figure 3F*). Together, these observations provide strong support for the idea that that orogastric gavage did not routinely lead to tissue damage that enables *S.* Typhimurium entry into the bloodstream. Instead, these data buttress the idea that *S.* Typhimurium disseminates from the intestine to extraintestinal organs early in the infection, sometime after oral inoculation and prior to significant intestinal replication.

To confirm our hypothesis that *S.* Typhimurium disseminates from the intestine to extraintestinal organs early in the infection, we sacrificed untreated mice 5 hr after orogastric inoculation with $\sim 10^8$ S. Typhimurium. Even at this early timepoint, $\sim 100$ *S.* Typhimurium CFU were found in liver samples (*Figure 3G*), indicating that the pathogen transits out of the intestine to the liver within 5 hr. At this point, there were also $\sim 100$ founders present in the liver samples, suggesting that little, if any, replication had occurred ($\sim 1$ CFU per founder). Notably, the number of liver founders at 5 hr was similar to the number of founders observed at 5 days (120 hr, *Figure 3H*), consistent with the idea that dissemination of *S.* Typhimurium from the intestine to the liver establishes the pathogen population that subsequently undergoes considerable replication. Furthermore, the barcodes present in liver samples were not present in any other samples. Together, these data suggest the initial rapid dissemination from the intestine seeds the majority of clones in the liver that are observed at later time points. In contrast to the liver, there were more founders present in samples from the intestine (particularly in the colon) at 5 hr versus 120 hr (*Figure 3H*). These data likely indicate that many of the founders observed in the intestine at 5 hr are shed in the feces prior to establishing a replicative niche, and demonstrates that the forces restricting the *S.* Typhimurium population in the intestine act over a period of >5 hr.

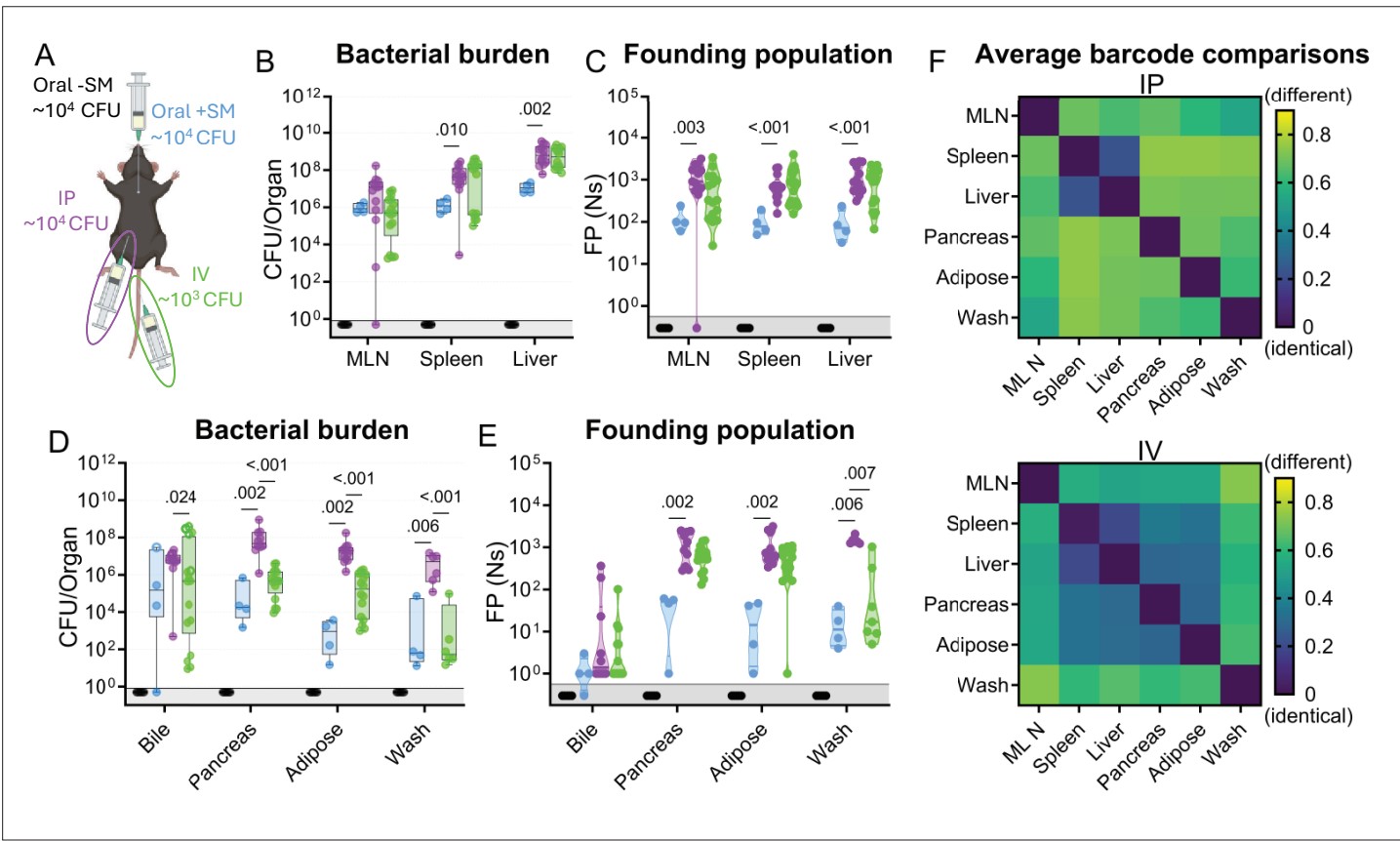

**Figure 4.** *S.* Typhimurium population dynamics following different routes of inoculation. (**A**) Infections were performed via 3 routes: orogastric gavage with (Oral +SM) or without (Oral -SM) streptomycin pretreatment, intraperitoneal injection (IP), and intravenous injection (IV). (**B–C**) Bacterial burden (**B**) and founding populations (**C**) in immune-rich extraintestinal organs. (**D–E**) Bacterial burden (**D**) and founding population (**E**) in extraintestinal samples. (**F**) Heatmaps of average genetic distance between extraintestinal organs after IP (top) and IV (bottom) inoculation. Box and whisker plots represent max-to-min and interquartile ranges. Truncated violin plots with all points displayed. Oral -SM n=5, Oral +SM n=4, IP n=16 (8 male, 8 female), IV n=16 (8 male, 8 female). Sex-disaggregated data in *Figure 4—figure supplement 1*. Mann-Whitney tests were used for statistical analyses. Values with p<0.1 shown. Abbreviations: MLN, mesenteric lymph node.

The online version of this article includes the following figure supplement(s) for figure 4:

**Figure supplement 1.** Sex-disaggregated data in mice after inoculation with $10^3$–$10^4$ CFU *S.*Typhimurium through different routes.

**Figure supplement 2.** Pathology observed in mouse adipose tissue.

## The primary bottleneck to *S.* Typhimurium extraintestinal infection occurs within or while the pathogen exits the intestine

The net bottleneck to *S.* Typhimurium spreading to extraintestinal organs following oral inoculation is a combination of individual bottlenecks, including barriers the pathogen encounters within the GI tract, escaping from the intestine, traveling to extraintestinal organs, and establishing a niche at extraintestinal sites. The proportional contribution of each of these bottlenecks is unknown. It is possible that they play equal roles or that one constitutes the majority of the net bottleneck. To further probe how the size of the founding population in extraintestinal organs is impacted by the bottlenecks *S.* Typhimurium experiences within or while exiting the intestine, we bypassed the intestinal bottleneck by administering the barcoded *S.* Typhimurium library intraperitoneally (IP) and intravenously (IV; *Figure 4A*), routes that have been used to model the pathogen's interactions with the immune system in extraintestinal infections (*Worley, 2023*; *Grant et al., 2008*). Lower doses of $10^4$ (IP) and $10^3$ (IV) CFU were used to avoid rapid death (*Xu and Hsu, 1992*). In addition, infections via orogastric gavage were performed at a dose of $10^4$ CFU both with and without streptomycin pretreatment to facilitate direct comparison between inoculation routes (*Figure 4A*).

Without streptomycin treatment at this lower dose, orogastric gavage (Oral -SM) did not lead to extraintestinal infection. Furthermore, there was no detectable *S.* Typhimurium (and consequently zero founders) in all the organs sampled (*Figure 4B–C*), consistent with a bottleneck that completely eliminated the $10^4$ cells in the inoculum either before *S.* Typhimurium escaped from the intestine or after spread from the intestine but prior to establishing a niche at extraintestinal sites. Moreover, these untreated mice did not exhibit any signs of disease, including diarrhea, lethargy, or weight loss (*Figure 4—figure supplement 1*). Thus, the net bottleneck to extraintestinal dissemination following orogastric inoculation results in less than 1 founder after inoculation of $10^4$ CFU.

The pathogen burden and number of founders in immune-rich organs were highly similar following IV and IP inoculation (*Figure 4B–C*). Furthermore, the numbers of founders in these organs were similar to the size of the inoculum, suggesting that *S.* Typhimurium does not experience a significant bottleneck when colonizing immune-rich organs following IV and IP inoculation. In streptomycin-treated animals given $10^4$ CFU by orogastric gavage, the pathogen burden and number of founders observed in immune-rich organs were ~10-fold lower than after IP and IV inoculation. These observations suggest that the net bottleneck that *S.* Typhimurium experiences while spreading from oral inoculation to extraintestinal immune-rich organs is tighter than the net bottleneck following either IV or IP inoculation. This additional ~10-fold constriction of the net bottleneck is likely contributed by the additional bottlenecks encountered by the pathogen in its route from oral inoculation to arrival in systemic organs, including escaping from the intestine.

Remarkably, even though most animals had a high *S.* Typhimurium burden in bile extracted from the gallbladder (*Figure 4D*), the number of founders was generally very low. Of animals with pathogen in the bile, more than half after all inoculation routes contained only a single founder (*Figure 4E*). The high frequency of clonality in the bile indicates the presence of a very tight bottleneck to colonization of the gallbladder. The factors that impede *S.* Typhimurium colonization of the gallbladder and bile are not clear, but once a single or a few bacterial cells take hold, they can replicate to yield a very high burden, with on average ~$3 \times 10^6$ (IP) and ~$9 \times 10^4$ (IV) CFU. Collectively, these observations support the hypothesis that the most restrictive bottleneck to *S.* Typhimurium dissemination occurs within or while the pathogen exits the intestine. However, spread to other organs, such as the gallbladder, are impeded by additional bottlenecks.

## Route of inoculation affects *S.* Typhimurium replication within and sharing among extraintestinal tissues

After IV and IP injection of *S.* Typhimurium there were similar burdens (*Figure 4B*) and founding population sizes (*Figure 4C*) in immune-rich organs. However, these routes yielded divergent outcomes at other sites. In the pancreas, perigonadal adipose tissue, and peritoneal washes, *S.* Typhimurium burdens were 100–1000-fold greater following IP versus IV inoculation (*Figure 4D*). Furthermore, at these sites, there was a trend toward more founders after IP inoculation (*Figure 4E*), although this trend may be explained by the 10-fold higher IP dose. Along with the increased pathogen burden, white lesions were also observed in the adipose tissue of many IP-inoculated and some IV-inoculated mice (*Figure 4—figure supplement 2*). We speculate that the increased *S.* Typhimurium burden may

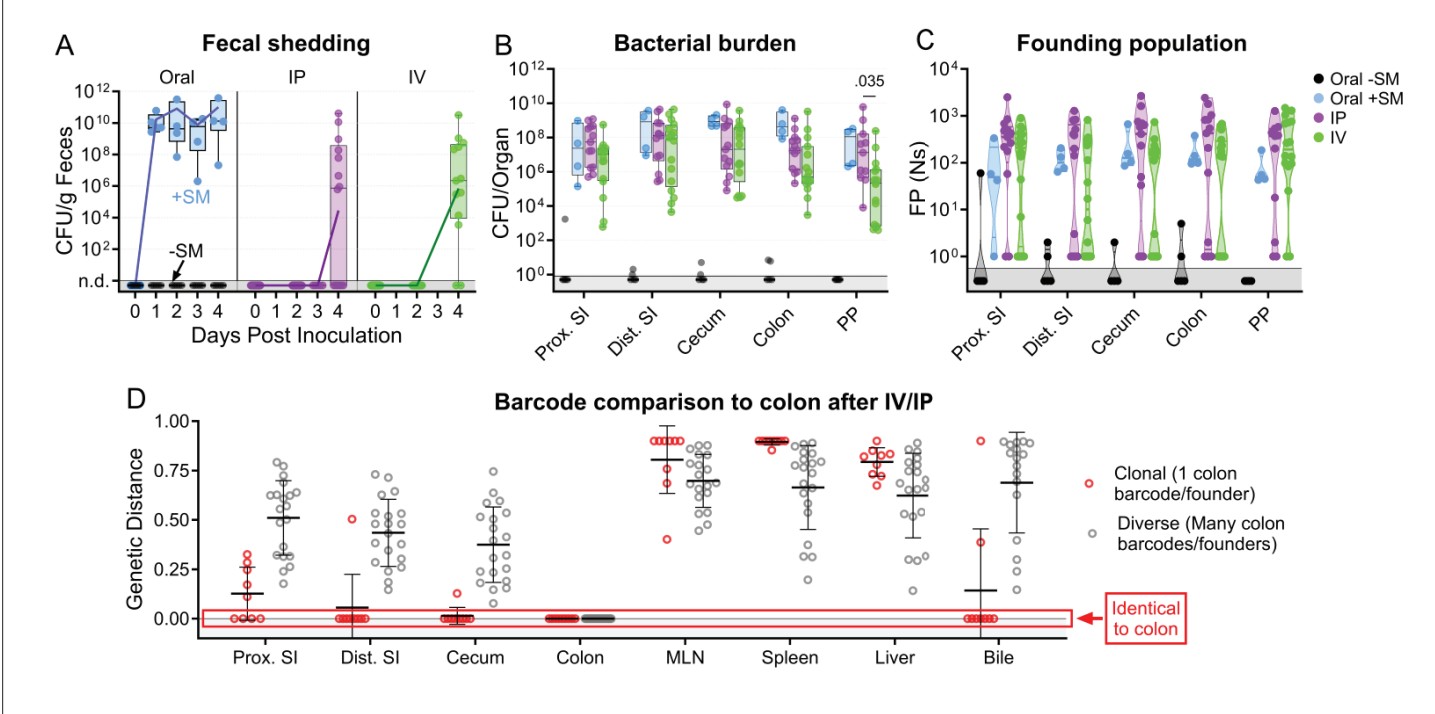

**Figure 5.** *S.* Typhimurium population dynamics after IV and IP inoculation differ compared to those after oral gavage. (**A**) Fecal shedding after orogastric gavage with (Oral +SM) or without (Oral -SM) streptomycin pretreatment, intraperitoneal injection (IP), and intravenous injection (IV) with *S.* Typhimurium. (**B**) Bacterial burdens in the intestine. Box and whisker plots represent max-to-min and interquartile ranges. (**C**) Founding population of GI organs. (**D**) Genetic distance comparison of colon samples of IV and IP animals to other sites disaggregated by number of colon founders. Means and standard deviations displayed. Mann-Whitney tests were used for statistical analyses. Values with p<0.1 shown. Abbreviations: MLN, mesenteric lymph node; PP, Peyer's patches; SI, small intestine.

originate from replication in the lesions. Exclusively after IP inoculation, there was a considerably higher pathogen burden in these three extraintestinal samples (pancreas, perigonadal adipose, and peritoneal wash) than either the size of the inoculum or founding population, suggesting an increase in localized replication in the peritoneal cavity, the site of inoculation, and adjacent tissue.

Barcode comparisons revealed that there was a consistently high level of sharing between the spleen and liver after IP and IV inoculation of *S.* Typhimurium (Avg. GD <0.25, ***Figure 4F***), indicating the spleen and liver are within the same compartment. Previous studies suggest sharing between the liver and spleen is initially minimal but increases over time (***Lam and Monack, 2014***; ***Grant et al., 2008***), consistent with early dissemination followed by sharing within the compartment as the infection progresses. After IP inoculation, however, the other extraintestinal sites were dissimilar to the liver/spleen compartment and each other, suggesting they represent separate compartments (GD >0.5, ***Figure 4F***, top). In contrast, following IV inoculation there was greater similarity (Avg. GD <0.5, ***Figure 4F***, bottom) between the *S.* Typhimurium populations in the liver/spleen compartment and those from the pancreas and adipose tissue (***Figure 4F***, more dark blue squares in the IV versus IP). Thus, these analyses of barcoded bacteria unveil how the route of inoculation can result in divergent population dynamics.

### *S.* Typhimurium can re-seed the intestine through the bile

Orogastric gavage of 10⁴ *S.* Typhimurium in streptomycin-treated animals resulted in high-level shedding (10¹⁰ CFU/g) by 1 day after inoculation, likely reflecting a rapid blooming of the *S.* Typhimurium population in the intestine. In marked contrast, *S.* Typhimurium shedding was not observed until 4 days after IV or IP pathogen inoculation; at this point, fecal shedding was detected in 8/16 and 11/16 animals inoculated IP or IV, respectively (***Figure 5A***). In addition, the *S.* Typhimurium burden in intestinal samples was generally higher in streptomycin-treated animals inoculated by the orogastric route versus untreated animals receiving an IV or IP inoculum (***Figure 5B***). The presence of *S.* Typhimurium

in the intestine and fecal shedding after systemic inoculation indicates that there are route(s) for *S.* Typhimurium to spread back into the intestine from extraintestinal organs. However, the absence of shedding in some animals and the marked delay in shedding after IP and IV relative to orogastric inoculation suggests that the *S.* Typhimurium population encounters substantial bottleneck(s) on the route(s) from extraintestinal sites back to the intestine.

Notably, the number of founders in intestinal samples from animals inoculated IP or IV had a bimodal distribution (*Figure 5C*). In some animals, IV and IP inoculation resulted in very few or even only a single founder in intestinal samples ('clonal' group, n=9), indicating that the population that re-seeded the intestine had been affected by a severely restrictive bottleneck (*Figure 5C*). The remaining animals in the IV and IP groups with colon colonization (n=22) had a much larger number of founders, approaching the number observed in the immune-rich samples (*Figure 4C*), suggestive of a highly permissive bottleneck. Together, these data strongly suggest that there are at least two routes by which extraintestinal *S.* Typhimurium can return to the intestine; one route has little, if any, bottleneck and gives rise to a diverse population in the intestine and the other route includes a very tight bottleneck and often results in a clonal intestinal population.

Comparisons of the genetic distances showed that the IV and IP groups with clonal intestinal populations had the same clone (barcode) throughout their respective GI tracts (clonal colons GD = 0, *Figure 5D*). In contrast, colons with diverse populations had less similarity to other regions of the intestine. Comparisons of the barcode similarity of clonal colon samples to extraintestinal samples provided insight into the likely extraintestinal source of the intestinal clone. 7 of the 9 animals with clonal populations in the colon had bile samples with genetic distances of zero, indicating the presence of the same clone and strongly suggesting that the bile clone seeded the intestine through the common bile duct. However, even though most bile samples containing *S.* Typhimurium from IP/IV inoculated mice were clonal (14/26) or pauciclonal (5/26 with Ns = 2–10), most intestinal samples were not clonal, indicating that bile is often not the dominant extraintestinal site that seeds the intestine. Furthermore, clonal bile did not necessarily give rise to a dominant intestinal clone (*Figure 6A*, animals 1 and 2). Interestingly, the bottleneck that characterizes the more permissive route to seeding the intestine from extraintestinal sites was even more permissive than the bottleneck that the pathogen experiences following orogastric inoculation (i.e. Ns values for the permissive IP and IV groups were generally greater than those for the orogastric group, *Figure 5C*). We speculate that the re-seeding route leading to the diverse *S.* Typhimurium population in the intestine represents a reversal of known mechanisms the pathogen employs to escape from the intestine (*Silva-García et al., 2019*).

## *S.* Typhimurium re-seeding the intestine through the bile correlates with gallbladder pathology

Unexpectedly, we observed that animals with clonal colon populations often had clonal bile populations as well as visible biliary pathology, such as darkening, cloudiness, or hardening (*Figure 6A–B*). We noticed that animals with diverse colon populations often had gallbladders and bile lacking any visible pathology, regardless of the bile's clonality. This led us to further analyze the association of gallbladder/bile pathology with the bile-driven intestinal re-seeding pathway. Overall, there was a small chance for any gallbladder/bile phenotype in all the infection schemes we tested, but there was a lower rate of bile colonization in untreated orogastrically gavaged animals (n=2; *Figure 6—figure supplement 1*).

Analysis of aggregated data from all infection routes and doses showed that increased pathogen burden in bile correlated with cloudy or hardened biliary pathology (*Figure 6D*), but not darkened bile. Additionally, loss of biliary epithelial cells and marked cellular infiltrates were observed in cloudy and hardened bile samples (*Figure 6C*). Thus, cloudy and hardened biliary pathology either enables the growth of *S.* Typhimurium in the bile or, more likely, is the result of *S.* Typhimurium growth in the bile. Estimates of the fraction of barcodes that contribute to similarity between the bile and colon (FRD, see Materials and methods) revealed that a greater fraction of bile barcodes was shared with the colon in animals with cloudy and hardened bile samples compared to bile samples without pathology (*Figure 6D*). Collectively, these observations show that hardened or cloudy biliary pathology reflects robust *S.* Typhimurium replication in the gallbladder and is correlated with seeding of the intestine. Indeed, fecal shedding correlates with bile burden in animals with hardened or cloudy biliary pathology, but not with animals whose bile was darkened or lacked pathology (*Figure 6F*). Thus,

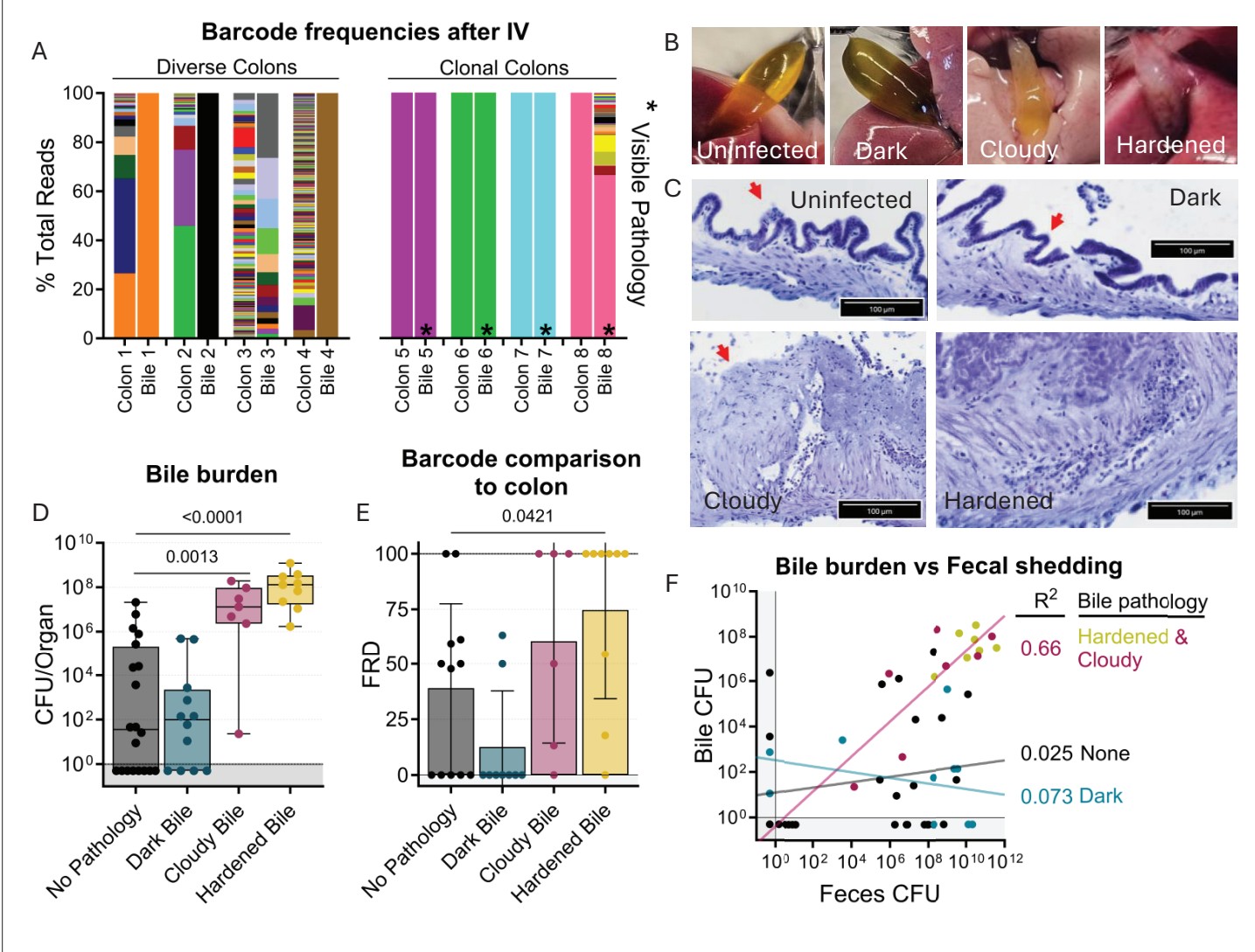

**Figure 6.** Bile re-seeding the intestine is correlated with gallbladder pathology. (**A**) Barcode frequency of clonal and diverse colons and bile in mice after IV inoculation. Asterisks indicate visible pathology in the bile. (**B**) Gross anatomy and (**C**) H&E-stained sections of normal, dark, cloudy, and hardened bile after *S*. Typhimurium infection. Red arrows indicate the luminal side of the gallbladder. (**D**) CFU in bile after all infection routes disaggregated by bile phenotype. Box and whisker plots represent max-to-min and interquartile ranges. (**E**) FRD of bile barcodes also found in colon samples. Bars are means with standard deviations. (**F**) Fecal shedding correlates with bile burden when hardened or cloudy bile is present. Line of best fit displayed. Mann-Whitney tests were used for statistical analyses. Values with p<0.1 shown.

The online version of this article includes the following figure supplement(s) for figure 6:

**Figure supplement 1.** Bile pathology correlates with the bile re-seeding pathway.

STAMPR-based analyses uncovered an unexpected link between gallbladder pathology and bile-related intestinal re-seeding.

## Discussion

*S*. Typhimurium is often used to model typhoidal disease in mice because it is highly adept at dissemination from the intestine, with a toolbox of diverse mechanisms to reach and replicate in extraintestinal organs. Previous studies of *Salmonella* population dynamics have been limited in resolution and focused on sites highly colonized by the pathogen, such as the cecum, feces, MLN, spleen, and liver (**Kaiser et al., 2013**; **Kaiser et al., 2014**; **Dybowski et al., 2015**; **Lam and Monack, 2014**; **Grant et al., 2008**). We built upon these studies using a highly diverse library of barcoded bacteria

containing over 55,000 unique tags, empowering us to quantify the size of the founding populations (*Abel et al., 2015*) and increasing the resolution of our genetic distance calculations (*Dybowski et al., 2017*). Additionally, we sampled a larger range of organs to evaluate the different patterns of dissemination and population compartmentalization after multiple inoculation routes and doses. With this complex barcoded library and the STAMPR analytical framework, we were able to observe the compartmentalization of the *Salmonella* population in the intestine, that dissemination out of the intestine occurs rapidly, and at least two distinct pathways re-seed the intestine.

Previous studies have used a limited number of unique barcodes (<30) to quantify *Salmonella* population dynamics in mice. Here, we employ a library that is over 1000-fold more diverse than previously used. Although low diversity libraries can be used to measure bottlenecks and dissemination, higher diversity libraries are advantageous for several reasons. High diversity libraries increase the upper limit of founding population measurements. For example, a library containing only 10 barcodes has a resolution limit of ~200 founders, meaning it would be unable to distinguish between organs with founding populations larger than 200 (e.g. 1000 and 10,000), because both organs would likely contain all 10 barcodes. In contrast, a library with 10,000 barcodes can be used to distinguish between a bottleneck resulting in Ns = 1000 and Ns = 10,000, since these bottlenecks result in a different number of barcodes in output samples. Furthermore, high diversity libraries reduce the likelihood that two tissue samples share the same barcode(s) due to random chance, enabling more accurate quantification of bacterial dissemination. For example, our analysis distinguishing subpopulations of Peyer's patches (*Figure 2—figure supplement 2*) would not have been possible with <100 barcodes.

By evaluating the identity and frequency of barcoded *S.* Typhimurium populations at sites throughout the intestine, we observed the presence of separate compartments along the intestinal tract containing unique subpopulations after orogastric gavage of the pathogen. Streptomycin pretreatment reduced the compartmentalization of *Salmonella* populations in the intestine and correlated with increased barcode sharing between the intestine and shed bacteria. We propose two non-mutually exclusive mechanisms that could account for these observations: (1) an expansion of the replicative niche that produces fecal bacteria (e.g. increased luminal replication) and/or (2) increased escape of *S.* Typhimurium from niches that normally do not contribute to the fecal population (e.g. intracellular tissue-associated bacteria). Since streptomycin treatment has been shown to increase the CFU in the lumen of the intestine and increase fecal shedding with minimal effect on the number of bacteria able to invade the intestinal epithelium (*Barthel et al., 2003*), the former idea seems more likely to be the major contributor.

In addition to finding distinct populations within the intestine, we also observed that the populations in extraintestinal organs were different from the populations in the intestine. The lack of similarity between intestinal and extraintestinal populations provides additional support for the idea that *Salmonella* disseminates from the intestine prior to substantial replication. Early dissemination is also supported by our discovery that the number of founders in the liver does not increase from 5 hr to 5 days post-inoculation, whereas the founding population in the intestine constricts during this period.

Among the extraintestinal organs, the liver and spleen often shared similar populations regardless of inoculation route. However, the route of inoculation influenced the sharing between *S.* Typhimurium populations from different organs. The IV and IP routes of inoculation yielded distinct patterns of sharing between the perigonadal adipose tissues, pancreas, and peritoneal wash. These two routes of inoculation are often considered interchangeable because they result in overall similar colonization and disease progression, but we found they lead to divergent patterns of pathogen population sharing between organs. This discovery illustrates the potency of our approach to unveil the impacts of infection route on pathogen spread.

Although our analyses cannot unequivocally prove the mechanisms *S.* Typhimurium employs to reach extraintestinal sites, evidence of known mechanisms are present in the STAMPR data. For example, the *S.* Typhimurium populations in the liver and MLN of untreated mice were somewhat similar (Avg. GD = 0.556 ± 0.139, *Figure 3C*), suggesting the lymphatic system, a well-characterized method of dissemination (*Li, 2022*; *Kaiser et al., 2013*; *Bravo-Blas et al., 2019*), was at least one of the mechanisms used to reach the liver. Interestingly, streptomycin-treatment caused a ~10-fold increase in founders in the liver and spleen (*Figure 3A*), indicating a relaxed net bottleneck to dissemination. Concurrently, in streptomycin treated animals, the *S.* Typhimurium population in the liver became more dissimilar from the populations in the MLN (Avg. GD = 0.760 ± 0.056, *Figure 3C*),

suggesting that the increase in founders reaching disseminated organs did not correlate with replication of these clones in the lymphatic system. Streptomycin treatment is known to increase intestinal inflammation, immune cell infiltration, and permeability of the intestinal epithelium (*Ray et al., 2022*; *Strati et al., 2021*; *Bazett et al., 2016*), and the pathogen may take advantage of these indirect effects of streptomycin treatment to disseminate more efficiently, resulting in increased founders at extraintestinal sites (*Worley, 2023*; *Li, 2022*; *Watson and Holden, 2010*; *Gogoi et al., 2019*).

As an enteric pathogen spread through the feces, the most expedient method for *Salmonella* propagation and transmission would be for the pathogen to enter the intestine, replicate within the intestine, and be shed into the feces. However, as we have observed, the microbiota creates a significant bottleneck, constricting the inoculum ~$10^6$-fold (*Figure 2B*), and preventing the majority of the intestinal population from shedding into the feces (*Figure 2E*). We propose that dissemination into extraintestinal organs, where *Salmonella* can replicate without competition with the microbiome, is an evolutionary boon for the pathogen. However, it creates a new challenge to bacterial transmission – re-entering the intestine to be shed in the feces. Using IP and IV inoculation, we were able to identify at least two routes by which *Salmonella* re-seeds the intestine. First, diverse populations can re-seed the intestine presumably by reversing known mechanisms of exit from the intestine (e.g. *Salmonella* infiltration through damaged intestinal epithelium outside of or within immune cells *Li, 2022*). Second, a highly bottlenecked population can re-enter the intestine through the common bile duct.

The bile re-seeding pathway was correlated with cloudy or hardened biliary pathology, increased clonality in the colon, and very high bile pathogen burden. Previous studies have also observed that increased intestinal clonality in *Salmonella* populations at the peak of infection is correlated with increased shedding in a different strain of mice (*Lam and Monack, 2014*), potentially due to bile re-seeding. It is tempting to speculate that the extremely high pathogen burden in the gallbladder in these mice accounts for both the gallbladder/bile pathology as well as the predilection for the bile clone to become dominant in the intestine. However, it is also possible that *S*. Typhimurium derived from a diseased biliary system has an enhanced capacity to become dominant in the intestine. Indeed, *Salmonella* has been previously shown to adapt to growth in bile through regulation of quorum sensing and virulence genes (*Yang et al., 2023*; *Tsai et al., 2020*; *Johnson et al., 2018*). Notably, in experimental *Listeria monocytogenes* (*Zhang et al., 2017*; *Chevée et al., 2024*) and *Psuedomonas auerginosa* (*Bachta et al., 2020*) infections in mice, clonal pathogen populations that replicate to a high burden in bile are also observed and routinely become the dominant clone in the intestine and feces. However, for all three pathogens, the mechanisms that underlie the highly restrictive gallbladder bottleneck remain unknown.

Long-term murine infections with *Salmonella* result in bile duct inflammation (*Pragasam et al., 2023*; *Dowling, 2000*), but to our knowledge no evidence of biliary inflammation after acute infection has been reported. *Salmonella* is also known to colonize gallbladder epithelia or the surface of gallstones in humans, often the source of pathogen in persistent *Salmonella* infections, and gallstone colonization has been shown to increase the risk of gallbladder cancer (*Crawford et al., 2010*; *Shrout, 2012*; *Tsai et al., 2020*; *Pragasam et al., 2023*). Of note, both *Salmonella* infection and gallstone formation in humans have increased risk in adult females in comparison to males (*Peer et al., 2021*; *Novacek, 2006*; *Dias et al., 2022*). In our experiments, we observed a moderate increase in *Salmonella* disease markers, such as diarrhea, weight loss, and fecal shedding, in female mice in comparison to males (*Figure 3—figure supplement 2*; *Figure 4—figure supplement 1*). Moreover, 9/10 mice with hardened bile were female, suggesting a strong correlation of this phenotype with sex and point to a sex bias in the murine model as well.

The use of highly complex barcoded libraries sharpens analytic resolution to the point where pathogen spread within each individual animal is easily distinguishable and can uncover unexpected and potentially significant insights into pathogen population dynamics during infection. For example, we observed up to 1000-fold variation in the number of *S*. Typhimurium founders in much of the intestine in orogastrically gavaged untreated animals (*Figure 2B*), even though the mice were genetically identical, came from the same vendor, and often littermates. This variation was present between mice but not within mice, with individual mice having consistent founders (either high or low) throughout the GI tract. Investigating the mechanisms accounting for this dramatic inter-animal variation is possible with our approach and is of interest because such marked differences in bottlenecks can determine whether pathogen exposure leads to infection. In conclusion, the use of the STAMPR framework

coupled with our highly diverse barcoded *S.* Typhimurium library has deepened our understanding of *Salmonella*'s highly interconnected and multidirectional dissemination cycle.

# Materials and methods

## Key resources table

| Reagent type (species) or resource | Designation | Source or reference | Identifiers | Additional information |
|---|---|---|---|---|
| Strain, strain background (*Mus musculus*) | C57BL/6 J | Jackson laboratory | Strain #:000664 RRID:IMSR_JAX:000664 | Mice |
| Strain, strain background (*Salmonella enterica* serovar Typhimurium) | SL1344 | *Hoiseth and Stocker, 1981* | NCBI:txid216597 | Bacterial strains |
| Software, algorithm | STAMPR scripts | *Hullahalli et al., 2021*; *Hullahalli, 2024* | https://github.com/hullahalli/stampr_rtisan | STAMP sample processing |

## Mice

8-week-old female and male C57BL/6 J mice were purchased from Jackson laboratory. Mice were acclimatized in the biosafety level 2 (BSL2) facility at the Brigham and Women's Hospital for at least 72 hr before use. The facility is temperature (68–75$^0$F) and humidity (50%) controlled with 12 hr light/dark cycles. All experiments involving mice were performed according to protocols reviewed and approved by the Brigham and Women's Hospital Institutional Animal Care and Use Committee (protocol 2016N000416) and in compliance with the Guide for the Care and Use of Laboratory Animals.

## Bacterial strains

A streptomycin-resistant strain of *Salmonella enterica* serovar Typhimurium SL1344 was used to create the barcoded library used here. Unless otherwise noted, *S.* Typhimurium was grown at 37 °C in LB broth or solid agar. As needed, media was supplemented with streptomycin (SM, 200 µg/ml) and/or kanamycin (KM, 50 µg/ml).

## STAMP library generation

The barcoded library was created using the pSM1 plasmid donor library, which contains >70,000 barcodes, and helper pJMP1339, which contains the Tn7 transposon and conjugation/replication machinery, as described (*Hullahalli et al., 2021*; *Abel et al., 2015*; *Holmes et al., 2025*; *Zhang et al., 2017*; *Chevée et al., 2024*; *Bachta et al., 2020*; *Campbell et al., 2023*). *S.* Typhimurium was heat shocked for 4 hr at 42 °C. A triparental conjugation was then used to introduce the pSM1 library into *S.* Typhimurium and transconjugants were selected for as SM- and KM-resistant colonies. Transconjugant colonies were pooled in PBS +20% glycerol and stored in aliquots at –80 °C. Illumina sequencing indicated the library contains ~55,000 unique barcodes with even distribution (*Figure 2—figure supplement 1*). The growth of the library in LB supplemented with SM was not significantly different than the parent strain (*Figure 2—figure supplement 1*). The resolution limit of STAMPR analysis with our library was ~700,000 founders; this was demonstrated through creation of a standard curve, comparing the true founding population (CFU following plating of a serially diluted culture) versus the calculated founding population (*Figure 2—figure supplement 1*).

## Mouse infection

The barcoded *S.* Typhimurium library was prepared by resuspending a frozen aliquot in LB and expanding for 3–5 hr at 37 °C with shaking unless otherwise noted. OD$_{600}$ was used to estimate the size of bacterial population in the culture before use. Following growth, bacteria were pelleted and resuspended in 1 x sterile PBS to the appropriate concentration (orogastric gavage 10$^9$ or 10$^5$ CFU/ml; oral via drinking 5x10$^9$ CFU/ml; IP 10$^5$ CFU/ml; IV 10$^4$ CFU/ml) and stored at room temperature prior to inoculation.

Streptomycin sulfate (100 µl of 200 mg/ml, United States Pharmacopeia grade) was given by orogastric gavage 24 hr before gavage with bacteria. Mice were deprived of food for 2–4 hr before orogastric gavage and given light sedation with isoflurane inhalation immediately prior to gavage. 100 µl inocula were gavaged into the stomach using a 1 ml syringe and sterile 18 G, 1.5-inch, 2 mm ball, flexible feeding needles (Braintree Scientific). Before oral inoculation via drinking, mice were deprived of food for 8 hr. 20 µl inocula were pipetted into the mouth of scruffed mice, who were then observed to ensure consumption. For IP infections, mice were restrained by a scruff, then a 1 ml syringe with a sterile 27 G, 1-inch needle was inserted into the lower right quadrant of the animal's abdomen at a –45 degree angle, and 100 µl inoculum was injected. Before IV inoculation, mice were placed into small containers on a heating pad to help promote tail vein dilation and restrained with a Broome-style restrainer (Plas Labs). 100 µl inocula were injected into the lateral tail vein using a sterile 27 G needle.

Bacterial counts were enumerated for all inoculums using serial dilutions on LB agar supplemented with SM. Mice were monitored daily for signs of infection including weight loss, fecal shedding, and diarrhea. Mice with weight loss of <5%, indicative of lack of disease, were excluded from further analysis (orogastric gavage -SM after $10^8$ CFU n=2, IP n=3).

## Necropsy

At the peak of the disease (4- or 5 days post-inoculation), at 5 hr post-inoculation, or as needed per the humane endpoint in our animal protocol, mice were sacrificed by isoflurane overdose followed by cervical dislocation. Peritoneal washes were obtained by injecting 5 ml sterile PBS into the peritoneal cavity of the animal with a 27 G needle and clamping the hole shut before gently massaging the abdomen for ~30 s before aspirating the wash from the peritoneal cavity. When possible, bile was obtained from the gallbladder with a 31 G short-needle insulin syringe (Sol-M, VWR) and aspirated into 90 µl sterile PBS. The gallbladder was then removed before harvesting the liver and other organs. If bile was not obtainable, the whole gallbladder was placed into a 1.5 ml tube containing 1 ml sterile PBS and 2x3.3 mm stainless steel balls and processed. All other organs (perigonadal adipose tissue, cecum, colon, distal third and proximal third of the small intestine, Peyer's patches, mesenteric lymph nodes, pancreas, liver, and spleen) were removed and individually placed into 1.5 ml tubes containing 1 ml sterile PBS (5 ml tube and 4 ml PBS for liver) and 2x3.3 mm stainless steel balls and then homogenized using a bead beater (Biospec Products). CFU of all organs was enumerated by serial dilution plating on LB agar containing SM. The remaining sample homogenate was plated on large LB agar plates containing SM for processing before sequencing to enumerate barcode frequency.

## Histology

The tissue samples were fixed in 4% paraformaldehyde in PBS and dehydrated in 20% sucrose in PBS. The dehydrated tissue was then embedded in OCT compound and sectioned to 10 µm. The sectioned slides were stained with hematoxylin and eosin (Abcam, #ab245880), washed with distilled water, and captured using Olympus Slideview VS200 (Rodent Histopathology Core, Dana-Farber/Harvard Cancer Center).

## STAMP sample processing

Samples were prepared for sequencing as described (*Hullahalli et al., 2021*; *Abel et al., 2015*; *Holmes et al., 2025*; *Zhang et al., 2017*; *Chevée et al., 2024*; *Bachta et al., 2020*; *Campbell et al., 2023*). Briefly, bacteria from large (150 mm x 15 mm) plates were resuspended in PBS +20% glycerol and stored at –80 °C. The frozen suspensions were diluted in water and boiled at 95 °C for 15 min to obtain genomic DNA. An Eppendorf epMotion 5075 liquid handling robot was used to multiplex primer composition for sample preparation before PCR amplification (primers listed in *Table 1*). OneTaq HS Quick-Load (New England Biolabs) was used to amplify the genomic barcodes by PCR for 25 cycles. The presence of amplicons was confirmed by agarose gel electrophoresis. Samples were pooled and purified using a Qiagen DNA cleanup kit. Then DNA concentrations were checked with a Qubit fluorimeter and sequenced with an Illumina NextSeq 1000/2000. FASTQ files were generated by Illumina's proprietary pipeline in BaseSpace with DRAGEN BCL Conver v3.10.4. Sequencing reads were then demultiplexed, mapped to the donor barcode library pSM1, and trimmed in R (version 4.3.0) using custom scripts to obtain counts for each barcode (Barcode counts are listed in Source data file 1). The

**Table 1.** Primer sequences for PCR.

| Name | Sequence |
| --- | --- |
| **Forward Primers** | |
| var21 | AATGATACGGCGACCACCGAGATCTACAC TCTTTCCCTACACGACGCTCTTCCGATCTA ATGATGGGTTAAAAAGGATCGATCC |
| var22 | AATGATACGGCGACCACCGAGATCTACACTC TTTCCCTACACGACGCTCTTCCGATCTAT GCGATGGGTTAAAAAGGATCGATCC |
| var23 | AATGATACGGCGACCACCGAGATCTA CACTCTTTCCCTACACGACGCTCTTCC GATCTTGCACGATGGGTTAAAAAGGATCGATCC |
| var24 | AATGATACGGCGACCACCGAGATCT ACACTCTTTCCCTACACGACGCTCTTCCGA TCTTCATTCGATGGGTTAAAAAGGATCGATCC |
| var25 | AATGATACGGCGACCACCGAGATCTACACTCT TTCCCTACACGACGCTCTTCCGATCTGAAT CGAGATGGGTTAAAAAGGATCGATCC |
| var26 | AATGATACGGCGACCACCGAGATCTACACTC TTTCCCTACACGACGCTCTTCCGATCTGTCAA CTTGATGGGTTAAAAAGGATCGATCC |
| var27 | AATGATACGGCGACCACCGAGATCTACAC TCTTTCCCTACACGACGCTCTTCCGATCT CGGCGTGGCGATGGGTTAAAAAGGATCGATCC |
| var28 | AATGATACGGCGACCACCGAGATCTACAC TCTTTCCCTACACGACGCTCTTCCGATCTCC TGTACCTTGATGGGTTAAAAAGGATCGATCC |
| **Reverse Primers** | |
| AD001 | CAAGCAGAAGACGGCATACGAGATCGT GATGTGACTGGAGTTCAGACGTGTGCTC TTCCGATCAGATCCTTGGCGGCAAGAAA |
| AD002 | CAAGCAGAAGACGGCATACGAGATACAT CGGTGACTGGAGTTCAGACGTGTGCTCT TCCGATCAGATCCTTGGCGGCAAGAAA |
| AD003 | CAAGCAGAAGACGGCATACGAGATGCC TAAGTGACTGGAGTTCAGACGTGTGCTC TTCCGATCAGATCCTTGGCGGCAAGAAA |
| AD004 | CAAGCAGAAGACGGCATACGAGATTG GTCAGTGACTGGAGTTCAGACGTGTG CTCTTCCGATCAGATCCTTGGCGGCAAGAAA |
| AD005 | CAAGCAGAAGACGGCATACGAGATC ACTGTGTGACTGGAGTTCAGACGTG TGCTCTTCCGATCAGATCCTTGGCGGCAAGAAA |
| AD006 | CAAGCAGAAGACGGCATACGAGATAT TGGCGTGACTGGAGTTCAGACGTGTGC TCTTCCGATCAGATCCTTGGCGGCAAGAAA |
| AD007 | CAAGCAGAAGACGGCATACGAGATGATCTGGT GACTGGAGTTCAGACGTGTGCTCTTCCGAT CAGATCCTTGGCGGCAAGAAA |
| AD008 | CAAGCAGAAGACGGCATACGAGATT CAAGTGTGACTGGAGTTCAGACGTG TGCTCTTCCGATCAGATCCTTGGCGGCAAGAAA |
| AD009 | CAAGCAGAAGACGGCATACGA GATCTGATCGTGACTGGAGTTCAGACGT GTGCTCTTCCGATCAGATCCTTGGCGGCAAGA AA |

*Table 1 continued on next page*

*Table 1 continued*

| Name | Sequence |
| --- | --- |
| AD010 | CAAGCAGAAGACGGCATACGAGATAAGCTAGTGACTGGAGTTCAGACGTGTGCTCTTCCGATCAGATCCTTGGCGGCAAGAAA |
| AD011 | CAAGCAGAAGACGGCATACGAGATGTAGCCGTGACTGGAGTTCAGACGTGTGCTCTTCCGATCAGATCCTTGGCGGCAAGAAA |
| AD012 | CAAGCAGAAGACGGCATACGAGATTACAAGGTGACTGGAGTTCAGACGTGTGCTCTTCCGATCAGATCCTTGGCGGCAAGAAA |
| AD013 | CAAGCAGAAGACGGCATACGAGATTTGACTGTGACTGGAGTTCAGACGTGTGCTCTTCCGATCAGATCCTTGGCGGCAAGAA |
| AD014 | CAAGCAGAAGACGGCATACGAGATGGAACTGTGACTGGAGTTCAGACGTGTGCTCTTCCGATCAGATCCTTGGCGGCAAGAAA |
| AD015 | CAAGCAGAAGACGGCATACGAGATTGACATGTGACTGGAGTTCAGACGTGTGCTCTTCCGATCAGATCCTTGGCGGCAAGAAA |
| AD016 | CAAGCAGAAGACGGCATACGAGATGGACGGGTGACTGGAGTTCAGACGTGTGCTCTTCCGATCAGATCCTTGGCGGCAAGAAA |
| AD018 | CAAGCAGAAGACGGCATACGAGATGCGGACGTGACTGGAGTTCAGACGTGTGCTCTTCCGATCAGATCCTTGGCGGCAAGAAA |
| AD019 | CAAGCAGAAGACGGCATACGAGATTTTCACGTGACTGGAGTTCAGACGTGTGCTCTTCCGATCAGATCCTTGGCGGCAAGAAA |
| AD020 | CAAGCAGAAGACGGCATACGAGATGGCCACGTGACTGGAGTTCAGACGTGTGCTCTTCCGATCAGATCCTTGGCGGCAAGAAA |
| AD021 | CAAGCAGAAGACGGCATACGAGATCGAAACGTGACTGGAGTTCAGACGTGTGCTCTTCCGATCAGATCCTTGGCGGCAAGAAA |
| AD022 | CAAGCAGAAGACGGCATACGAGATCGTACGGTGACTGGAGTTCAGACGTGTGCTCTTCCGATCAGATCCTTGGCGGCAAGAAA |
| AD023 | CAAGCAGAAGACGGCATACGAGATCCACTCGTGACTGGAGTTCAGACGTGTGCTCTTCCGATCAGATCCTTGGCGGCAAGAAA |
| AD025 | CAAGCAGAAGACGGCATACGAGATATCAGTGTGACTGGAGTTCAGACGTGTGCTCTTCCGATCAGATCCTTGGCGGCAAGAAA |
| AD027 | CAAGCAGAAGACGGCATACGAGATAGGAATGTGACTGGAGTTCAGACGTGTGCTCTTCCGATCAGATCCTTGGCGGCAAGAA |

founding population (Ns) was determined through comparison to the frequencies of barcodes in the undiluted library using the STAMPR pipeline (*Hullahalli et al., 2021*). Cavalli-Sforza chord distance (2√2/π=0.9) was used to compare the genetic distance of samples from the same animal. FRD was calculated between sample A and sample B as $FRD_{A-B} = \frac{ln(RD_{A-B})}{ln(number\ of\ unique\ barcodes\ in\ sample\ B)}$. $RD_{A-B}$ is a measure for the number of shared barcodes that contribute to genetic similarity (defined as GD <0.8) between two samples, described extensively in *Hubbard et al., 2019*.

## Statistical analysis

Statistical analyses were performed using GraphPad Prism version 10.1.2. Information regarding the number of samples and statistical tests are described in the figure legends. Geometric means, geometric standard deviations, and non-parametric tests were used for analyzing bacterial burden and founding population data. Means, standard deviations, and non-parametric tests were used for comparisons of animal weights, genetic distances, and barcode identity. As our lower limit of detection was near 1, we substituted null values with a value between 0.1 and 0.8 for graphical representation.

## Acknowledgements

The authors would like to thank the Waldor lab for their insight and helpful conversations.

This work was funded by the Howard Hughes Medical Institute (MKW), grants from the National Institutes of Health R01 AI042347 (MKW), P30 DK034854 (IWC), and fellowships from the National Institutes of Health T32 DK007477-37 (IWC), F31 AI156949 (KH).

We thank Dana-Farber/Harvard Cancer Center in Boston, MA, for the use of the Rodent Histopathology Core, which provided microscopy services. Dana-Farber/Harvard Cancer Center is supported in part by a NCI Cancer Center Support Grant # NIH 5 P30 CA06516.

This article is subject to HHMI's Open Access to Publications policy. HHMI lab heads have previously granted a nonexclusive CC BY 4.0 license to the public and a sublicensable license to HHMI in their research articles. Pursuant to those licenses, the author-accepted manuscript of this article can be made freely available under a CC BY 4.0 license immediately upon publication.

---

## Additional information

### Funding

| Funder | Grant reference number | Author |
| --- | --- | --- |
| National Institute of Allergy and Infectious Diseases | R01 AI042347 | Matthew K Waldor |
| National Institute of Diabetes and Digestive and Kidney Diseases | P30 DK034854 | Ian W Campbell |
| National Institute of Allergy and Infectious Diseases | F31 AI156949 | Karthik Hullahalli |
| National Institute of Diabetes and Digestive and Kidney Diseases | T32 DK007477-37 | Ian W Campbell |

The funders had no role in study design, data collection and interpretation, or the decision to submit the work for publication.

### Author contributions

Julia A Hotinger, Conceptualization, Data curation, Formal analysis, Validation, Investigation, Visualization, Methodology, Writing – original draft, Writing – review and editing; Ian W Campbell, Conceptualization, Data curation, Visualization, Methodology, Writing – original draft, Writing – review and editing; Karthik Hullahalli, Conceptualization, Software, Formal analysis, Methodology; Akina Osaki, Conceptualization, Formal analysis, Investigation; Matthew K Waldor, Conceptualization, Supervision, Funding acquisition, Writing – original draft, Writing – review and editing

### Author ORCIDs

Ian W Campbell  http://orcid.org/0000-0003-3019-2560
Karthik Hullahalli  https://orcid.org/0000-0003-3064-2090
Matthew K Waldor  https://orcid.org/0000-0003-1843-7000

---

### Ethics

All experiments involving mice were performed according to protocols reviewed and approved by the Brigham and Women's Hospital Institutional Animal Care and Use Committee (protocol 2016N000416) and in compliance with the Guide for the Care and Use of Laboratory Animals.

Reviewer #1 (Public review): https://doi.org/10.7554/eLife.101388.3.sa1

Reviewer #2 (Public review): https://doi.org/10.7554/eLife.101388.3.sa2

Author response https://doi.org/10.7554/eLife.101388.3.sa3

---

## Additional files

### Supplementary files

MDAR checklist

Source data 1. Barcode counts for all samples in the manuscript (separate file).

### Data availability

STAMPR scripts are available in Github (copy archived at *Hullahalli, 2024*). Dataset S1 contains the barcode counts used for STAMPR analysis. Requests for further information, resources, or reagents will be fulfilled by the Lead Contact, Matthew K. Waldor (mwaldor@research.bwh.harvard.edu).

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
