## [Editor Report · eLife Assessment]

This **important** study reports a detailed quantification of the population dynamics of *Salmonella enterica* serovar Typhimurium in mice. Bacterial burden and founding population sizes across various organs were quantified, revealing pathways of dissemination and reseeding of the gastrointestinal tract from systemic organs. Using various techniques, including genetic distance measurements, the authors present **compelling** evidence to support their conclusions, thus presenting new knowledge that will be of broad interest to scientists focusing on infectious diseases.

---

## [Referee Report · Reviewer #1 (Public review)]

Hotinger et al. explore the population dynamics of *Salmonella enterica* serovar Typhimurium in mice using genetically tagged bacteria. In addition to physiological observations, pathology assessments, and CFU measurements, the study emphasizes quantifying host bottleneck sizes that limit *Salmonella* colonization and dissemination. The authors also investigate the genetic distances between bacterial populations at various infection sites within the host.

Initially, the study confirms that pretreatment with the antibiotic streptomycin before inoculation via orogastric gavage increases the bacterial burden in the gastrointestinal (GI) tract, leading to more severe symptoms and heightened fecal shedding of bacteria. This pretreatment also significantly reduces between-animal variation in bacterial burden and fecal shedding. The authors then calculate founding population sizes across different organs, discovering a severe bottleneck in the intestine, with founding populations reduced by approximately 10^6-fold compared to the inoculum size. Streptomycin pretreatment increases the founding population size and bacterial replication in the GI tract. Moreover, by calculating genetic distances between populations, the authors demonstrate that, in untreated mice, *Salmonella* populations within the GI tract are genetically dissimilar, suggesting limited exchange between colonization sites. In contrast, streptomycin pretreatment reduces genetic distances, indicating increased exchange.

In extraintestinal organs, the bacterial burden is generally not substantially increased by streptomycin pretreatment, with significant differences observed only in the mesenteric lymph nodes and bile. However, the founding population sizes in these organs are increased. By comparing genetic distances between organs, the authors provide evidence that subpopulations colonizing extraintestinal organs diverge early after infection from those in the GI tract. This hypothesis is further tested by measuring bacterial burden and founding population sizes in the liver and GI tract at 5 and 120 hours post-infection. Additionally, they compare orogastric gavage infection with the less injurious method of infection via drinking, finding similar results for CFUs, founding populations, and genetic distances. These results argue against injuries during gavage as a route of direct infection.

To bypass bottlenecks associated with the GI tract, the authors compare intravenous (IV) and intraperitoneal (IP) routes of infection. They find approximately a 10-fold increase in bacterial burden and founding population size in immune-rich organs with IV/IP routes compared to orogastric gavage in streptomycin-pretreated animals. This difference is interpreted as a result of "extra steps required to reach systemic organs."

While IP and IV routes yield similar results in immune-rich organs, IP infections lead to higher bacterial burdens in nearby sites, such as the pancreas, adipose tissue, and intraperitoneal wash, as well as somewhat increased founding population sizes. The authors correlate these findings with the presence of white lesions in adipose tissue. Genetic distance comparisons reveal that, apart from the spleen and liver, IP infections lead to genetically distinct populations in infected organs, whereas IV infections generally result in higher genetic similarity.

Finally, the authors investigate GI tract reseeding, identifying two distinct routes. They observe that the GI tracts of IP/IV-infected mice are colonized either by a clonal or a diversely tagged bacterial population. In clonally reseeded animals, the genetic distance within the GI tract is very low (often zero) compared to the bile population, which is predominantly clonal or pauciclonal. These animals also display pathological signs, such as cloudy/hardened bile and increased bacterial burden, leading the authors to conclude that the GI tract was reseeded by bacteria from the gallbladder bile. In contrast, animals reseeded by more complex bacterial populations show that bile contributes only a minor fraction of the tags. Given the large founding population size in these animals' GI tracts, which is larger than in orogastrically infected animals, the authors suggest a highly permissive second reseeding route, largely independent of bile. They speculate that this route may involve a reversal of known mechanisms that the pathogen uses to escape from the intestine.

The manuscript presents a substantial body of work that offers a meticulously detailed understanding of the population dynamics of S. Typhimurium in mice. It quantifies the processes shaping the within-host dynamics of this pathogen and provides new insights into its spread, including previously unrecognized dissemination routes. The methodology is appropriate and carefully executed, and the manuscript is well-written, clearly presented, and concise. The authors' conclusions are well-supported by experimental results and thoroughly discussed. This work underscores the power of using highly diverse barcoded pathogens to uncover the within-host population dynamics of infections and will likely inspire further investigations into the molecular mechanisms underlying the bottlenecks and dissemination routes described here.

---

## [Referee Report · Reviewer #2 (Public review)]

In this paper, Hotinger et. al. propose an improved barcoded library system, called STAMPR, to study *Salmonella* population dynamics during infection. Using this system, the authors demonstrate significant diversity in the colonization of different *Salmonella* clones (defined by the presence of different barcodes) not only across different organs (liver, spleen, adipose tissues, pancreas and gall bladder) but also within different compartments of the same gastrointestinal tissue. Additionally, this system revealed that microbiota competition is the major bottleneck in *Salmonella* intestinal colonization, which can be mitigated by streptomycin treatment. However, this has been demonstrated previously in numerous publications. They also show that there was minimal sharing between populations found in the intestine and those in the other organs. Upon IV and IP infection to bypass the intestinal bottleneck, they were able to demonstrate, using this library, that *Salmonella* can renter the intestine through two possible routes. One route is essentially the reverse path used to escape the gut, leading to a diverse intestinal population; while the other, through the bile, typically results in a clonal population.

Comments on latest version:

The authors have addressed my concerns.

---

## [Author Response]

The following is the authors’ response to the original reviews.

**Public Reviews:**

**Reviewer #1 (Public review):**
Hotinger et al. explore the population dynamics of *Salmonella enterica* serovar Typhimurium in mice using genetically tagged bacteria. In addition to physiological observations, pathology assessments, and CFU measurements, the study emphasizes quantifying host bottleneck sizes that limit *Salmonella* colonization and dissemination. The authors also investigate the genetic distances between bacterial populations at various infection sites within the host.Initially, the study confirms that pretreatment with the antibiotic streptomycin before inoculation via orogastric gavage increases the bacterial burden in the gastrointestinal (GI) tract, leading to more severe symptoms and heightened fecal shedding of bacteria. This pretreatment also significantly reduces between-animal variation in bacterial burden and fecal shedding. The authors then calculate founding population sizes across different organs, discovering a severe bottleneck in the intestine, with founding populations reduced by approximately 10^6-fold compared to the inoculum size. Streptomycin pretreatment increases the founding population size and bacterial replication in the GI tract. Moreover, by calculating genetic distances between populations, the authors demonstrate that, in untreated mice, *Salmonella* populations within the GI tract are genetically dissimilar, suggesting limited exchange between colonization sites. In contrast, streptomycin pretreatment reduces genetic distances, indicating increased exchange.In extraintestinal organs, the bacterial burden is generally not substantially increased by streptomycin pretreatment, with significant differences observed only in the mesenteric lymph nodes and bile. However, the founding population sizes in these organs are increased. By comparing genetic distances between organs, the authors provide evidence that subpopulations colonizing extraintestinal organs diverge early after infection from those in the GI tract. This hypothesis is further tested by measuring bacterial burden and founding population sizes in the liver and GI tract at 5 and 120 hours post-infection. Additionally, they compare orogastric gavage infection with the less injurious method of infection via drinking, finding similar results for CFUs, founding populations, and genetic distances. These results argue against injuries during gavage as a route of direct infection.To bypass bottlenecks associated with the GI tract, the authors compare intravenous (IV) and intraperitoneal (IP) routes of infection. They find approximately a 10-fold increase in bacterial burden and founding population size in immune-rich organs with IV/IP routes compared to orogastric gavage in streptomycin-pretreated animals. This difference is interpreted as a result of "extra steps required to reach systemic organs."While IP and IV routes yield similar results in immune-rich organs, IP infections lead to higher bacterial burdens in nearby sites, such as the pancreas, adipose tissue, and intraperitoneal wash, as well as somewhat increased founding population sizes. The authors correlate these findings with the presence of white lesions in adipose tissue. Genetic distance comparisons reveal that, apart from the spleen and liver, IP infections lead to genetically distinct populations in infected organs, whereas IV infections generally result in higher genetic similarity.Finally, the authors investigate GI tract reseeding, identifying two distinct routes. They observe that the GI tracts of IP/IV-infected mice are colonized either by a clonal or a diversely tagged bacterial population. In clonally reseeded animals, the genetic distance within the GI tract is very low (often zero) compared to the bile population, which is predominantly clonal or pauciclonal. These animals also display pathological signs, such as cloudy/hardened bile and increased bacterial burden, leading the authors to conclude that the GI tract was reseeded by bacteria from the gallbladder bile. In contrast, animals reseeded by more complex bacterial populations show that bile contributes only a minor fraction of the tags. Given the large founding population size in these animals' GI tracts, which is larger than in orogastrically infected animals, the authors suggest a highly permissive second reseeding route, largely independent of bile. They speculate that this route may involve a reversal of known mechanisms that the pathogen uses to escape from the intestine.The manuscript presents a substantial body of work that offers a meticulously detailed understanding of the population dynamics of S. Typhimurium in mice. It quantifies the processes shaping the within-host dynamics of this pathogen and provides new insights into its spread, including previously unrecognized dissemination routes. The methodology is appropriate and carefully executed, and the manuscript is well-written, clearly presented, and concise. The authors' conclusions are well-supported by experimental results and thoroughly discussed. This work underscores the power of using highly diverse barcoded pathogens to uncover the within-host population dynamics of infections and will likely inspire further investigations into the molecular mechanisms underlying the bottlenecks and dissemination routes described here.Major point:Substantial conclusions in the manuscript rely on genetic distance measurements using the Cavalli-Sforza chord distance. However, it is unclear whether these genetic distance measurements are independent of the founding population size. I would anticipate that in populations with larger founding population sizes, where the relative tag frequencies are closer to those in the inoculum, the genetic distances would appear smaller compared to populations with smaller founding sizes independent of their actual relatedness. This potential dependency could have implications for the interpretation of findings, such as those in Figures 2B and 2D, where antibiotic-pretreated animals consistently exhibit higher founding population sizes and smaller genetic distances compared to untreated animals.

Thank you for raising this important point regarding reliance on cord distances for gauging genetic distance in barcoded populations. The reviewer is correct that samples with more founders will be more similar to the inoculum and thus inherently more similar to other samples that also have more founders. However, creation of libraries containing very large numbers of unique barcodes can often circumvent this issue. In this case, the effect size of chance-based similarity is not large enough to change the interpretation of the data in Figures 2B and 2D. In our case, the library has ~6x10^4^ barcodes, and the founding populations in Figure 2B are ~10^3^. Randomly resampling to create two populations of 10^3^ cells from an initial population with 6x10^4^ barcodes is expected to yield largely distinct populations with very little similarity. Thus, the similarity between streptomycin-treated populations in Figure 2D is likely the result of biology rather than chance.

**Reviewer #2 (Public review):**
In this paper, Hotinger et. al. propose an improved barcoded library system, called STAMPR, to study *Salmonella* population dynamics during infection. Using this system, the authors demonstrate significant diversity in the colonization of different *Salmonella* clones (defined by the presence of different barcodes) not only across different organs (liver, spleen, adipose tissues, pancreas, and gall bladder) but also within different compartments of the same gastrointestinal tissue. Additionally, this system revealed that microbiota competition is the major bottleneck in *Salmonella* intestinal colonization, which can be mitigated by streptomycin treatment. However, this has been demonstrated previously in numerous publications. They also show that there was minimal sharing between populations found in the intestine and those in the other organs. Upon IV and IP infection to bypass the intestinal bottleneck, they were able to demonstrate, using this library, that Salmonella can renter the intestine through two possible routes. One route is essentially the reverse path used to escape the gut, leading to a diverse intestinal population; while the other, through the bile, typically results in a clonal population. Although the authors showed that the STAMPR pipeline improved the ability to identify founder populations and their diversity within the same animal during infections, some of the conclusions appear speculative and not fully supported.(1) It's particularly interesting how the authors, using this system, demonstrate the dominant role of the microbiota bottleneck in *Salmonella* colonization and how it is widened by antibiotic treatment (Figure 1). Additionally, the ability to track *Salmonella* reseeding of the gut from other organs starting with IV and IP injections of the pathogen provides a new tool to study population dynamics (Figure 5). However, I don't think it is possible to argue that the proximal and distal small intestine, Peyer's patches (PPs), cecum, colon, and feces have different founder populations for reasons other than stochastic variations. All the barcoded *Salmonella* clones have the same fitness and the fact that some are found or expanded in one region of the gastrointestinal tract rather than another likely results from random chance - such as being forced in a specific region of the gut for physical or spatial reasons-and subsequent expansion, rather than any inherent biological cause. For example, some bacteria may randomly adhere to the mucus, some may swim toward the epithelial layer, while others remain in the lumen; all will proliferate in those respective sites. In this way, different founder populations arise based on random localization during movement through the gastrointestinal tract, which is an observation, but it doesn't significantly contribute to understanding pathogen colonization dynamics or pathogenesis. Therefore, I would suggest placing less emphasis on describing these differences or better discussing this aspect, especially in the context of the gastrointestinal tract.

Thank you for helping us identify this area for further clarification. We agree with the reviewer’s interpretation that seeding of proximal and distal small intestine, Peyer's patches (PPs), cecum, colon, and feces with different founder populations is likely caused by stochastic variations, consistent with separate stochastic bottlenecks to establishing these separate niches. To clarify this point we have modified the text in the results section, “Streptomycin treatment decreases compartmentalization of *S*. Typhimurium populations within the intestine”.

Change to text:

“Except for the cecum and colon, in untreated animals the *S*. Typhimurium populations in different regions of the intestine were dissimilar (Avg. GD ranged from 0.369 to 0.729, 2D left); i.e., there is little sharing between populations in the intestine. These data suggest that there are separate bottlenecks in different regions of the intestine that cause stochastic differences in the identity of the founders. Interestingly, when these founders replicate, they do not mix, remaining compartmentalized with little sharing between populations throughout the intestinal tract (i.e., barcodes found in one region are not in other regions, Figure S3). This was surprising as the luminal contents, an environment presumably conducive to bacterial movement, were not removed from these samples.”

In this section we are interested in the underlying biology that occurs after the initial bottleneck to preserve this compartmentalization during outgrowth of the intestinal population. In other words, what prevents these separate populations from merging (e.g., what prevents the bacteria replicating in the proximal small intestine from traveling through the intestine and establishing a niche in the distal small intestine)? While we do not explore the mechanisms of compartmentalization, we observe that it is disrupted by streptomycin pretreatment, suggesting a microbiota-dependent biological cause.

(2) I do think that STAMPR is useful for studying the dynamics of pathogen spread to organs where *Salmonella* likely resides intracellularly (Figure 3). The observation that the liver is colonized by an early intestinal population, which continues to proliferate at a steady rate throughout the infection, is very interesting and may be due to the unique nature of the organ compared to the mucosal environment. What is the biological relevance during infection? Do the authors observe the same pattern (Figures 3C and G) when normalizing the population data for the spleen and mesenteric lymph nodes (mLN)? If not, what do the authors think is driving this different distribution?

Thank you for raising this interesting point. These data indicate that the liver is seeded from the intestine early during infection. The timing and source of dissemination have relevance for understanding how host and pathogen variables control the spread of bacteria to systemic sites. For example, our conclusion (early dissemination) indicates that the immune state of a host at the time of exposure to a pathogen, and for a short period thereafter, are what primarily influence the process of dissemination, not the later response to an active infection.

We observe that the liver and mucosal environments within the intestine have similar colonization behaviors. Both niches are seeded early during infection, followed by steady pathogen proliferation and compartmentalization that apparently inhibits further seeding. This results in the identity of barcodes in the liver population remaining distinct from the intestinal populations, and the intestinal populations remaining distinct from each other.

We observe a similar pattern to the liver in the spleen and MLN (the barcodes in the spleen and MLN are dissimilar to the population in the intestine). To clarify this point, we have modified the text (below) and added this analysis as a supplemental figure (S4).

Change to text:

Genetic distance comparison of liver samples to other sites revealed that, regardless of streptomycin treatment, there was very little sharing of barcodes between the intestine and extraintestinal sites (Avg. GD >0.75, Figure 3C). Furthermore, the MLN and spleen populations also lacked similarity with the intestine (Figure S4). These analyses strongly support the idea that S. Typhimurium disseminates to extraintestinal organs relatively early following inoculation, before it establishes a replicative niche in the intestine.

(3) Figure 6: Could the bile pathology be due to increased general bacterial translocation rather than *Salmonella* colonization specifically? Did the authors check for the presence of other bacteria (potentially also proliferating) in the bile? Do the authors know whether Salmonella's metabolic activity in the bile could be responsible for gallbladder pathology?

The reviewer raises interesting points for future work. We did not check whether other bacterial species are translocating during *S*. Typhimurium infection. The relevance of *Salmonella*’s metabolic activity is also very interesting, and we hope these questions will be answered by future studies.

**Recommendations for the authors:**

**Reviewer #1 (Recommendations for the authors):**
Minor points:(1) P. 9/10 "... the marked delay in shedding after IP and IV relative to orogastric inoculation suggest that the S. Typhimurium population encounters substantial bottleneck(s) on the route(s) from extraintestinal sites back to the intestine.": Can you conclude that from the data? It could also be possible that there is a biological mechanism (other than chance events) that delays the re-entry to the intestine.

We propose that the delay in shedding indicates additional obstacles that bacteria face when re-entering the intestine, and that there are likely biological mechanisms that cause this delay. However, these unknown mechanisms effectively act as additional bottlenecks by causing a stochastic loss of population diversity.

(2) P. 11 "...both organs would likely contain all 10 barcodes. In contrast, a library with 10,000 barcodes can be used to distinguish between a bottleneck resulting in Ns = 1,000 and Ns = 10,000, since these bottlenecks result in a different number of barcodes in output samples. Furthermore, high diversity libraries reduce the likelihood that two tissue samples share the same barcode(s) due to random chance, enabling more accurate quantification of bacterial dissemination.": I agree with the general analysis, but I find it misleading to talk about the presence of barcodes when the analyses in this manuscript are based on the much more powerful comparison of relative abundance of individual tags instead of their presence or absence.

The reviewer raises an excellent point, and the distinction between relative abundance versus presence/absence is discussed extensively in the original STAMPR manuscript. Although relative abundance is powerful, the primary metric used in this study (Ns) is calculated principally from the number of barcodes, corrected (via simulations) for the probability of observing the same barcode across distinct founders. Although this correction procedure does rely on barcode abundance, the primary driver of founding population quantification is the number of barcodes.

(3) P.14 "the library in LB supplemented with SM was not significantly different than the parent strain" and Figure 2C: How was significance tested? How many times were the growth curves recorded? On my print-out, the red color has different shades for different growth curves.

Significance was tested with a Mann-Whitney and growth curves were performed 5 times. Growth curves are displayed with 50% opacity, and as a result multiple curves directly on top of each other appear darker. The legend to S2 has been modified accordingly.

(4) P.16: close bracket in the equation for FRD calculation.

Done

(5) Figure 2C "Average CFU per founder": I found the wording confusing at first as I thought you divided the average bacterial burden per organ by Ns, instead of averaging the CFU/Ns calculated for each mouse.

The wording has been clarified.

(6) Figure 3B: It would be helpful to include expected genetic distances in the schematic as it is difficult to infer the genetic distance when only two of three, respectively, different "barcode colors" are used. While I find the explanation in the main text intuitive, a graphical representation would have helped me.

Thank you for the suggestion. Unfortunately, using colors to represent barcodes is imperfect and limits the diversity that can be depicted. We have modified Figure 3B to further clarify.

(7) Figure 3C: Why do you compare the genetic distance to the liver, when you discuss the genetic distance of the intestinal population? Is it not possible that the intestinal populations are similar to the extraintestinal organs except the liver?

For clarity, we chose to highlight exclusively the liver. However, we observed a similar pattern to the liver in other extraintestinal organs. To clarify the generalizability of this point we have added a supplemental figure with comparisons to MLN and Spleen (Supplemental figure S4) as well as further text.

(8) Figure 3C & S5A: I found "+SM" and "+SM, Drinking" confusing and would have preferred "+SM, Gavage" and "+SM, Drinking" for clarity.

Done, thank you for the suggestion.

(9) Figure 3G&H: I find it worthy of discussion that the bacterial burden increases over time, while the founding population decreases. Does that not indicate that replication only occurs at specific sites leading to the amplification of only a few barcodes and thereby a larger change of the relative barcode abundance compared to the inoculum?

From 5h to 120h the size of the founding population decreases in multiple intestinal sites. This likely indicates that the impact of the initial bottleneck is still ongoing at 5h, although further temporal analysis would be required to define the exact timing of the bottleneck. Notably, the passage time through the mouse intestine is ~5h. Many of the founders observed at 5h could be a population that will never establish a replicative niche, and failing to colonize be shed in the feces, bottlenecking the population between 5h and 120h. To clarify this point we have added the following text:

Section “S. Typhimurium disseminates out of the intestine before establishing an intestinal replicative niche”.

“In contrast to the liver, there were more founders present in samples from the intestine (particularly in the colon) at 5 hours versus 120 hours (Figure 3H). These data likely indicate that many of the founders observed in the intestine at 5 hours are shed in the feces prior to establishing a replicative niche, and demonstrates that the forces restricting the S. Typhimurium population in the intestine act over a period of > 5 hours.”

(10) Figure S2A: I do not understand this figure. Why are there more than 70.000 tags listed? I was under the impression the barcode library in S. Typhimurium had 55.000 tags while only the plasmid pSM1 had more than 70.000 (but the plasmid should not be relevant here). Why are there distinct lines at approximately 10^-5 and a bit lower? I would have expected continuously distributed barcode frequencies.

During barcode analysis, each library is mapped to the total barcode list in the barcode donor pSM1, which contains ~70,000 barcodes. This enables consistent analysis across different bacterial libraries. The designation “barcode number” refers to the barcode number in pSM1, meaning many of the barcodes in the *Salmonella* library are at zero reads. This graph type was chosen to show there was no bias toward a particular barcode, however there is significant overlap of the points, making individual barcode frequencies difficult to see. We have changed the x-axis to state “pSM1 Barcode Number” and clarified in the figure legend.

Since the y-axes on these graphs is on a log10 scale, the lines represent barcodes with 1 read, 2 reads, 3 reads, etc. As the number of reads per barcode increases linearly, the space between them decreases on logarithmic axes.

(11) There are a few typos in the figure legends of the supplementary material. For example Figure S2: S. Typhimurium not italicized, ~7x105 no superscript. Fig. S4&5 ", Open circles" is "O" is capitalized.

Typos have been corrected.